# Optimizing Data Augmentation through Bayesian Model Selection

**Madi Matymov**[*]
KAUST
Saudi Arabia

**Ba-Hien Tran**
Huawei Paris Research Center
France

**Michael Kampffmeyer**
UiT The Arctic University of Norway
Norwegian Computing Center, Norway

**Markus Heinonen**
Aalto University
Finland

**Maurizio Filippone**
KAUST
Saudi Arabia

## Abstract

Data Augmentation (DA) has become an essential tool to improve robustness and generalization of modern machine learning. However, when deciding on DA strategies it is critical to choose parameters carefully, and this can be a daunting task which is traditionally left to trial-and-error or expensive optimization based on validation performance. In this paper, we counter these limitations by proposing a novel framework for optimizing DA. In particular, we take a probabilistic view of DA, which leads to the interpretation of augmentation parameters as model (hyper)-parameters, and the optimization of the marginal likelihood with respect to these parameters as a Bayesian model selection problem. Due to its intractability, we derive a tractable Evidence Lower BOund (ELBO), which allows us to optimize augmentation parameters jointly with model parameters. We provide extensive theoretical results on variational approximation quality, generalization guarantees, invariance properties, and connections to empirical Bayes. Through experiments on computer vision and NLP tasks, we show that our approach improves calibration and yields robust performance over fixed or no augmentation. Our work provides a rigorous foundation for optimizing DA through Bayesian principles with significant potential for robust machine learning.

## 1 Introduction

Data Augmentation (DA) (Van Dyk & Meng, 2001) is an essential element behind the success of modern machine learning (see, e.g., Shorten & Khoshgoftaar, 2019, and references therein). In supervised learning, DA amounts to creating copies of the data in the training set, and perturbing these with sensible transformations that preserve label information. The success of DA is connected with the current trend of employing over-parameterized models based on neural networks, which require large amounts of data to be trained effectively (Alabdulmohsin et al., 2022). It has been shown that DA has strong connections with regularization (Zhang et al., 2017; Dao et al., 2019), and it can provide a better estimation of the expected risk (Shao et al., 2022; Chen et al., 2020; Lyle et al., 2020; Deng et al., 2022). Therefore, it is expected for DA to enhance generalization.

For a given problem, once transformation for DA are chosen, it is then necessary to decide on their parameters. For example, in image classification, if we choose to apply transformations in the form of rotations, what range of angles should we choose? Careful choices of DA parameters are important to obtain performance improvements. For example, in the case of rotations applied to the popular MNIST dataset, large rotation angles can turn a '9' into a '6', negatively impacting training. In the literature, DA parameters are often suggested after some trial-and-error. Direct optimization of DA parameters could also be approached via grid-search or Bayesian optimization by recording performance on a validation set, but this is very costly due to the need to perform a large number of training runs.

In this paper, we propose a novel approach to optimize DA which counters these limitations. In particular, we take a probabilistic view of DA, whereby we treat DA parameters as model (hyper-)

---

[*]Corresponding author: `madi.matymov@kaust.edu.sa`

Figure 1: **OPTIMA obtains the best calibration.** Example of ResNet-18 on CIFAR10; see details in Appendix F.1.

parameters. We then consider the optimization of such parameters as a Bayesian model selection problem. Due to the intractability of the Bayesian model selection objective (i.e., the marginal likelihood), we derive a tractable ELBO, which allows us to optimize DA parameters jointly with model parameters, bypassing the need to perform expensive cross-validation or grid search. We provide an extensive theoretical analysis, which indicates robust predictive performance and low Expected Calibration Error (ECE) as demonstrated by the experiments (see, e.g., Fig. 1). Our main contributions are as follows:

**Methodology:** We introduce OPTIMA (OPTImizimg Marginalized Augmentations), a novel framework to learn DA parameters grounded on Bayesian principles. We then provide a tractable variational approximation which allows for the optimization of both model parameters and DA parameters, yielding a practical and fast alternative to manual tuning or expensive black-box optimization of DA parameters.

**Theory:** We provide a comprehensive theoretical analysis, establishing a cohesive framework to understand our Bayesian approach to DA, highlighting its principled nature. Our analysis includes: (**i**) The analysis of the variational approximation's quality, guiding DA distribution design (§ 4.1). (**ii**) A derivation of PAC-Bayes generalization guarantees (§ 4.2) and demonstration on how OPTIMA promotes model invariance and smoother decision boundaries (§ 4.3). (**iii**) A demonstration of improved uncertainty quantification and calibration through proper marginalization over DA parameters (§ 4.4). (**iv**) The establishing of empirical Bayes optimality (§ 4.5) for data-driven DA strategies, complemented by information-theoretic insights (§ 4.6) into how learned DA enhances inference.

**Empirical Validation:** We support OPTIMA and the theoretical developments with rigorous empirical validation on various tasks (§ 5), including regression, image classification on standard benchmarks (e.g., CIFAR10 and IMAGENET), and an additional natural language classification task (SST-5). Across all these settings—spanning both continuous geometric transformations and discrete text perturbations—our experiments consistently demonstrate that OPTIMA improves generalization, model calibration, and robustness to out-of-distribution data compared to models trained with fixed or no augmentation strategies.

Overall, our work demonstrates how Bayesian principles, specifically through a (partial or full) variational treatment of both model and augmentation parameters, can be effectively leveraged to develop a practical, scalable, and principled framework for optimizing DA, moving beyond expensive trial-and-error or validation-based procedures for optimal DA.

## 2 BACKGROUND AND RELATED WORK

We consider supervised learning tasks, where mappings from inputs $\mathbf{x} \in \mathbb{R}^D$ to labels $\mathbf{y} \in \mathbb{R}^O$ are learned from $N$ training observations $\mathcal{D} = \{(\mathbf{x}_i, \mathbf{y}_i)\}_{i=1}^N$. A common approach is to find a loss minimizing point estimates, which is equivalent to maximizing a log likelihood $\log p(\mathbf{Y} \mid \boldsymbol{\theta}, \mathbf{X})$, where $\mathbf{X}$ and $\mathbf{Y}$ denote all inputs and labels, respectively.

**Marginal likelihood and ELBO.** In the Bayesian approach we choose a prior $p(\boldsymbol{\theta})$, and infer the posterior distributions over parameters and predictive distribution for a new data point $\mathbf{x}^\star$ as:

$$p(\boldsymbol{\theta} \mid \mathcal{D}) = \frac{p(\mathbf{Y} \mid \boldsymbol{\theta}, \mathbf{X})p(\boldsymbol{\theta})}{p(\mathbf{Y} \mid \mathbf{X})}, \qquad (1) \qquad p(\mathbf{y}^\star \mid \mathbf{x}^\star, \mathcal{D}) = \int p(\mathbf{y}^\star \mid \mathbf{x}^\star, \boldsymbol{\theta})p(\boldsymbol{\theta} \mid \mathcal{D})d\boldsymbol{\theta}. \qquad (2)$$

The denominator of Eq. 1 is the *marginal likelihood*, representing the data likelihood under the prior:

$$p(\mathbf{Y} \mid \mathbf{X}, \boldsymbol{\phi}) = \int p(\mathbf{Y} \mid \boldsymbol{\theta}, \mathbf{X}, \boldsymbol{\phi})p(\boldsymbol{\theta} \mid \boldsymbol{\phi})d\boldsymbol{\theta}, \qquad (3)$$

where we made explicit the dependence on continuous hyper-parameters $\boldsymbol{\phi}$. We can perform model selection by choosing the one with highest log-marginal likelihood, also known as *model evidence*. The intractability of this objective motivates us to employ variational inference to obtain a tractable lower bound to be optimized with respect to a parametric surrogate posterior $q(\boldsymbol{\theta})$,

$$\log p(\mathbf{Y} \mid \mathbf{X}, \boldsymbol{\phi}) \geq \mathbb{E}_{q(\boldsymbol{\theta})}\big[\log p(\mathbf{Y} \mid \boldsymbol{\theta}, \mathbf{X}, \boldsymbol{\phi})\big] - \mathrm{KL}\big[q(\boldsymbol{\theta}) \,\|\, p(\boldsymbol{\theta} \mid \boldsymbol{\phi})\big] \quad =: \text{ELBO} \qquad (4)$$

**Data augmentation in neural Networks.** In DA, we apply transformations $T_{\boldsymbol{\gamma}}(\mathbf{x})$ parameterized by $\boldsymbol{\gamma}$ to the inputs at training time. In image classification common transformations include rotations, translations, flips, and color manipulations, while in natural language popular augmentations involve word substitutions and syntactic transformations (Shorten & Khoshgoftaar, 2019; Feng et al., 2021). During training, for each sample in a mini-batch, we first sample a transformation parameter $\boldsymbol{\gamma}$ and then apply $T_{\boldsymbol{\gamma}}(\mathbf{x})$. This approach has proven highly effective in improving generalization in deep learning (Shorten & Khoshgoftaar, 2019).

**Augmentations overcount evidence.** Naïvely replicating augmented examples $\{(T_{\boldsymbol{\gamma}}(\mathbf{x}_i), \mathbf{y}_i)\}$ as if fully independent effectively multiplies the evidence (3), *overcounting* the likelihood (Wilson & Izmailov, 2020). For a single data point $(\mathbf{x}_i, \mathbf{y}_i)$, this yields a likelihood $\prod_{k=1}^{K} p(\mathbf{y}_i \mid T_{\boldsymbol{\gamma}_k}(\mathbf{x}_i), \boldsymbol{\theta})$, equivalent to raising $p(\mathbf{y}_i \mid \mathbf{x}_i, \boldsymbol{\theta})$ to the power $K$. This overcounting can artificially shrink posterior uncertainty and degrade calibration, undermining a key advantage of Bayesian methods.

## 2.1 RELATED WORKS

**Data augmentation optimization.** Optimizing data augmentation (DA) parameters has been approached via computationally expensive reinforcement learning (AutoAugment (Cubuk et al., 2019)), made more efficient by population-based training (Ho et al., 2019). Others formulate it as density matching, often using black-box search like Bayesian optimization (Snoek et al., 2012), or differentiable policy search (Lim et al., 2019; Cubuk et al., 2020; Hataya et al., 2020), or as gradient matching (Zheng et al., 2022). Bi-level optimization has also been used, but remains expensive and often relies on strong relaxations (Liu et al., 2021; Li et al., 2020; Hataya et al., 2022; Mounsaveng et al., 2021). These methods typically rely on heuristics and complex search pipelines. More recently, DA has been framed as invariance-constrained learning with regularized objectives (Benton et al., 2020) or via non-parametric models solved with costly Markov chain Monte Carlo (MCMC) (Hounie et al., 2023).

**Probabilistic perspectives of DA.** Probabilistic views of DA have shown perturbed inputs can induce degenerate (Izmailov et al., 2021) or tempered likelihoods (Kapoor et al., 2022), informing studies on DA's role in the cold-posterior effect (Wenzel et al., 2020; Bachmann et al., 2022). Nabarro et al. (2022) proposed an integral likelihood similar to ours using a Jensen lower bound, and Heinonen et al. (2025) defined an augmented likelihood via label smoothing and input mollification (Tran et al., 2023). Kapoor et al. (2022) analyzed augmentations through a Dirichlet likelihood. Related latent-variable formulations also appear in work such as Chen et al. (2020) and Chatzipantazis et al. (2023), which consider probabilistic transformations but do not optimize augmentation parameters within a joint Bayesian model. However, these approaches generally use fixed, unoptimized augmentation parameters. In contrast, Wang et al. (2023) modeled DA with stochastic output layers and auxiliary variables for MAP optimization via expectation maximization, while Wu & Williamson (2024) applied MixUp (Zhang et al., 2018) for martingale posteriors (Fong et al., 2023). Broader connections link DA to kernel methods for task-specific invariances (Dao et al., 2019), though not directly to Bayesian inference. More directly, van der Wilk et al. (2018) learned invariances via marginal likelihood for Gaussian processes (Williams & Rasmussen, 2006), an idea Immer et al. (2022) extended to BNNs (Neal, 1996; Tran et al., 2022) using the Laplace approximation (MacKay, 1992; Daxberger et al., 2021; Immer et al., 2021), but without theoretical generalization guarantees.

**PAC-Bayes generalization bounds.** PAC-Bayes bounds (McAllester, 1999; Catoni, 2007; Alquier, 2024) offer theoretical guarantees for Bayesian methods, including in deep learning (Dziugaite & Roy, 2017; Lotfi et al., 2022; Wilson, 2025). However, prior work rarely treats augmentation parameters as latent variables within this framework. We unify these directions by making the augmentation distribution a key component of the model's likelihood, deriving novel theoretical results that characterize its benefits.

## 3 AUGMENTATION OPTIMIZATION THROUGH BAYESIAN MODEL SELECTION

**Augmentation as Marginalization.** In this section we treat the optimization of DA parameters as Bayesian model selection. To do so, we start by defining a transformation-augmented likelihood:

$$p(\mathbf{y} \mid \mathbf{x}, \boldsymbol{\theta}, \boldsymbol{\phi}) = \mathbb{E}_{p(\boldsymbol{\gamma} \mid \boldsymbol{\phi})}\Big[p\big(\mathbf{y} \mid T_{\boldsymbol{\gamma}}(\mathbf{x}), \boldsymbol{\theta}\big)\Big], \tag{5}$$

where $T_{\boldsymbol{\gamma}}(\mathbf{x})$ is the transformed input under augmentation distribution $\boldsymbol{\gamma} \sim p(\boldsymbol{\gamma} \mid \boldsymbol{\phi})$ parameterized by $\boldsymbol{\phi}$. This formulation treats augmentation as marginalization over transformations rather than data replication. This method averages over transformations to contribute each original example exactly once, as opposed to the overcounting effect in the case of naïve augmentation. As we will see, this yields a more calibrated posterior with appropriate uncertainty quantification.

The data likelihood given model parameters $\boldsymbol{\theta}$ and augmentation parameters $\boldsymbol{\phi}$ is

$$p(\mathcal{D} \mid \boldsymbol{\theta}, \boldsymbol{\phi}) = \prod_{i=1}^{N} \mathbb{E}_{p(\boldsymbol{\gamma} \mid \boldsymbol{\phi})}\Big[p(\mathbf{y}_i \mid T_{\boldsymbol{\gamma}}(\mathbf{x}_i), \boldsymbol{\theta})\Big]. \tag{6}$$

Taking a fully Bayesian treatment, we assign a prior $p(\boldsymbol{\phi})$ on the augmentation parameters $\boldsymbol{\phi}$, making $\boldsymbol{\phi}$ a latent variable alongside $\boldsymbol{\theta}$. The joint distribution over all variables is $p(\mathcal{D}, \boldsymbol{\theta}, \boldsymbol{\phi}, \boldsymbol{\gamma}) = p(\boldsymbol{\theta})p(\boldsymbol{\phi})p(\boldsymbol{\gamma} \mid \boldsymbol{\phi})p(\mathcal{D}|\boldsymbol{\theta}, \boldsymbol{\phi})$. Our goal is to approximate the posterior $p(\boldsymbol{\theta}, \boldsymbol{\phi} \mid \mathcal{D})$, which is typically intractable. To address this challenge, we employ variational inference (Jordan et al., 1999).

**Augmented Evidence Lower Bound.** For variational inference, we introduce a variational distribution $q(\boldsymbol{\theta}, \boldsymbol{\phi}) = q(\boldsymbol{\theta})q(\boldsymbol{\phi})$ to approximate the posterior $p(\boldsymbol{\theta}, \boldsymbol{\phi} \mid \mathcal{D})$. The standard ELBO is a lower bound on the log marginal likelihood $\mathcal{L} := \log p(\mathcal{D}) = \log \iiint p(\mathcal{D}, \boldsymbol{\theta}, \boldsymbol{\phi}, \boldsymbol{\gamma}) \, d\boldsymbol{\theta} \, d\boldsymbol{\phi} \, d\boldsymbol{\gamma}$. Using Jensen's inequality with $q(\boldsymbol{\theta}, \boldsymbol{\phi})$ and with standard manipulations, we obtain the ELBO, which consists of a data-fitting term and two regularization terms $\text{KL}(q(\boldsymbol{\theta})\|p(\boldsymbol{\theta}))$ and $\text{KL}(q(\boldsymbol{\phi})\|p(\boldsymbol{\phi}))$:

$$\mathcal{L} \geq \mathbb{E}_{q(\boldsymbol{\theta})q(\boldsymbol{\phi})p(\boldsymbol{\gamma} \mid \boldsymbol{\phi})}\left[\sum_{i=1}^{N} \log p(\mathbf{y}_i \mid T_{\boldsymbol{\gamma}}(\mathbf{x}_i), \boldsymbol{\theta})\right] - \text{KL}(q(\boldsymbol{\theta})\|p(\boldsymbol{\theta})) - \text{KL}(q(\boldsymbol{\phi})\|p(\boldsymbol{\phi})). \tag{7}$$

**Optimization of the ELBO.** The augmented ELBO presented in Eq. 7 is optimized by jointly updating the parameters of the variational distributions $q(\boldsymbol{\theta})$ and $q(\boldsymbol{\phi})$ using stochastic gradient-based methods. This involves sampling from these distributions (often via reparameterization) and from the DA distribution $p(\boldsymbol{\gamma} \mid \boldsymbol{\phi})$ to compute Monte Carlo estimates of the expectation term, and then backpropagating through the objective. A detailed algorithm, specific choices for parameterizing $p(\boldsymbol{\gamma} \mid \boldsymbol{\phi})$ and $q(\boldsymbol{\phi})$ for continuous and discrete transformations, and other practical implementation aspects are discussed in Appendix D.

## 4 THEORETICAL ANALYSIS

We present a comprehensive analysis of the proposed DA approach based on Bayesian model selection, analyzing its properties from multiple perspectives: variational approximation quality, generalization guarantees, invariance properties, and connections to empirical Bayes. Our analysis includes a direct comparison with naïve DA, which amounts in treating augmented samples as training samples. This analysis provides a rigorous foundation for OPTIMA while yielding practical insights for implementation.

## 4.1 VARIATIONAL APPROXIMATION WITH AUGMENTATION

We begin by analyzing the quality of our variational approximation when incorporating DA.

**Proposition 4.1** (Jensen Gap Bound). *The augmentation distribution variance and model sensitivity control the Jensen gap introduced by our lower bound approximation. If $f(\boldsymbol{\gamma}) = \log p(\mathbf{y} \mid T_{\boldsymbol{\gamma}}(\mathbf{x}), \boldsymbol{\theta})$ is L-Lipschitz in $\boldsymbol{\gamma}$, and $\boldsymbol{\gamma} \sim p(\boldsymbol{\gamma}|\boldsymbol{\phi})$ is sub-Gaussian with variance proxy $\sigma^2$, then:*

$$\log \mathbb{E}_{\boldsymbol{\gamma}}\big[p(\mathbf{y} \mid T_{\boldsymbol{\gamma}}(\mathbf{x}), \boldsymbol{\theta})\big] - \mathbb{E}_{\boldsymbol{\gamma}}\big[\log p(\mathbf{y} \mid T_{\boldsymbol{\gamma}}(\mathbf{x}), \boldsymbol{\theta})\big] \leq \frac{L^2 \sigma^2}{2}. \tag{8}$$

*Also, this bound is tight when $f(\boldsymbol{\gamma})$ is approximately linear in the high-probability region of $p(\boldsymbol{\gamma}|\boldsymbol{\phi})$.*

The proof is presented in Appendix B.1. This result has important implications for optimizing the augmentation distribution $p(\boldsymbol{\gamma}|\boldsymbol{\phi})$:

> **Corollary 4.2** (Optimal Augmentation Variance). *The optimal variance $\sigma_{\boldsymbol{\phi}}^2$ for the augmentation distribution balances two competing factors:*
>
> *1. Increasing $\sigma_{\boldsymbol{\phi}}^2$ improves exploration of the augmentation space.*
>
> *2. Decreasing $\sigma_{\boldsymbol{\phi}}^2$ tightens the variational bound.*
>
> *For models with high sensitivity to augmentations (large L), smaller variance is preferred to maintain bound tightness.*

This corollary provides practical guidance for setting augmentation distribution parameters, suggesting that highly sensitive models benefit from more conservative augmentation strategies.

## 4.2 GENERALIZATION GUARANTEES

To analyze the generalization of our Bayesian-optimized DA, we leverage the PAC-Bayes framework (McAllester, 1999; Catoni, 2007); see Appendix C for a primer. PAC-Bayes theory provides high-probability upper bounds on the true risk (generalization error) of a learning algorithm that outputs a distribution over hypotheses (a "posterior"). These bounds typically depend on the empirical risk observed on the training data and a complexity term, often expressed as the KL divergence between this posterior and a data-independent prior distribution. By extending this framework to our setting, we can formally quantify how well the model with learned DA parameters will perform on unseen data. We first present a PAC-Bayes bound for OPTIMA, then provide a theorem that explicitly compares OPTIMA to naïve DA, demonstrating superior generalization.

**Definition 4.3** (True and Empirical Risks). Given the transformation function $T_{\boldsymbol{\gamma}}(\mathbf{x})$, we define:

- *True risk:* $R(\boldsymbol{\theta}, \boldsymbol{\phi}) = \mathbb{E}_{(\mathbf{x},\mathbf{y})\sim P}\big[-\log \mathbb{E}_{p(\boldsymbol{\gamma}|\boldsymbol{\phi})} p(\mathbf{y} \mid T_{\boldsymbol{\gamma}}(\mathbf{x}), \boldsymbol{\theta})\big]$.

- *Our empirical risk:* $\hat{R}(\boldsymbol{\theta}, \boldsymbol{\phi}) = -\frac{1}{N}\sum_{i=1}^{N} \log \mathbb{E}_{p(\boldsymbol{\gamma}|\boldsymbol{\phi})} p(\mathbf{y}_i \mid T_{\boldsymbol{\gamma}}(\mathbf{x}_i), \boldsymbol{\theta})$.

- *Empirical risk for naïve augmentation ($K$ samples per datapoint):*

$$\hat{R}_{\text{naïve}}(\boldsymbol{\theta}) = -\frac{1}{N}\frac{1}{K}\sum_{i=1}^{N}\sum_{k=1}^{K} \log p(y_i \mid T_{\boldsymbol{\gamma}_k}(x_i), \boldsymbol{\theta}), \quad \boldsymbol{\gamma}_k \sim p(\boldsymbol{\gamma} \mid \boldsymbol{\phi})$$

**Theorem 4.4** (PAC-Bayes with Augmented Likelihood). *For an i.i.d. dataset $\mathcal{D} = \{(\mathbf{x}_i, \mathbf{y}_i)\}_{i=1}^{N}$ drawn from an unknown distribution $P$, any prior $p(\boldsymbol{\theta}, \boldsymbol{\phi})$, and any posterior $q(\boldsymbol{\theta}, \boldsymbol{\phi}) = q(\boldsymbol{\theta})q(\boldsymbol{\phi})$ over the hypothesis space $\boldsymbol{\theta} \times \boldsymbol{\phi}$, with probability at least $1 - \delta$ over the draw of $\mathcal{D}$:*

$$\mathbb{E}_{q(\boldsymbol{\theta},\boldsymbol{\phi})}\big[R(\boldsymbol{\theta}, \boldsymbol{\phi})\big] \leq \mathbb{E}_{q(\boldsymbol{\theta},\boldsymbol{\phi})}\big[\hat{R}(\boldsymbol{\theta}, \boldsymbol{\phi})\big] + \sqrt{\frac{KL(q(\boldsymbol{\theta}, \boldsymbol{\phi})\|p(\boldsymbol{\theta}, \boldsymbol{\phi})) + \log\frac{2\sqrt{N}}{\delta}}{2N}}, \tag{9}$$

*where $KL(q(\boldsymbol{\theta}, \boldsymbol{\phi})\|p(\boldsymbol{\theta}, \boldsymbol{\phi})) = KL(q(\boldsymbol{\theta})\|p(\boldsymbol{\theta})) + KL(q(\boldsymbol{\phi})\|p(\boldsymbol{\phi}))$ if $p(\boldsymbol{\theta}, \boldsymbol{\phi}) = p(\boldsymbol{\theta})p(\boldsymbol{\phi})$.*

To explicitly demonstrate that OPTIMA generalizes better than naïve DA, we now compare the PAC-Bayes bounds of both methods, showing that OPTIMA yields a tighter bound due to proper marginalization over transformations. We encourage reader refer to Appendix C for further discussion.

**Remark.** As usual with Monte Carlo estimates, the naïve risk $\hat{R}_{\mathrm{naive}}$ is a consistent approximation of the true marginalization when $K$ is large enough, which is the setting assumed in Theorem 4.5.

**Theorem 4.5** (Generalization Advantage of Bayesian-Optimized Augmentation). *Consider a model parameterized by $\boldsymbol{\theta} \in \Theta$, and let $\boldsymbol{\phi} \in \Phi$ parameterize an augmentation distribution $p(\boldsymbol{\gamma} \mid \boldsymbol{\phi})$, where $\boldsymbol{\gamma}$ defines transformations $T_{\boldsymbol{\gamma}}(\mathbf{x})$.*

*We consider the following assumptions:*

1. *The transformation distribution $p(\boldsymbol{\gamma} \mid \boldsymbol{\phi})$ is such that $\mathbb{E}_{p(\boldsymbol{\gamma}|\boldsymbol{\phi})} p(\mathbf{y} \mid T_{\boldsymbol{\gamma}}(\mathbf{x}), \boldsymbol{\theta})$ can be computed or approximated accurately.*

2. *The variational posteriors $q(\boldsymbol{\theta}, \boldsymbol{\phi})$ and $q(\boldsymbol{\theta})$ are optimized to minimize their respective bounds.*

3. *The KL divergences $KL(q(\boldsymbol{\theta}, \boldsymbol{\phi}) \| p(\boldsymbol{\theta}, \boldsymbol{\phi}))$ and $KL(q(\boldsymbol{\theta}) \| p(\boldsymbol{\theta}))$ are comparable, i.e., the complexity penalties are similar.*

*Under these assumptions, the PAC-Bayes bound for OPTIMA is tighter than that for naïve DA:*

$$\mathbb{E}_{q(\boldsymbol{\theta}, \boldsymbol{\phi})}[R(\boldsymbol{\theta}, \boldsymbol{\phi})] \leq \mathbb{E}_{q(\boldsymbol{\theta})}[R(\boldsymbol{\theta})] - \Delta, \tag{10}$$

*where $\Delta = \mathbb{E}_{q(\boldsymbol{\theta}, \boldsymbol{\phi})} \left[ \frac{1}{N} \sum_{i=1}^{N} \Delta_{\boldsymbol{\phi}}(\mathbf{x}_i, \mathbf{y}_i) \right] \geq 0$, and $\Delta_{\boldsymbol{\phi}}(\mathbf{x}_i, \mathbf{y}_i) = \log \mathbb{E}_{p(\boldsymbol{\gamma}|\boldsymbol{\phi})} p(\mathbf{y}_i \mid T_{\boldsymbol{\gamma}}(\mathbf{x}_i), \boldsymbol{\theta}) - \mathbb{E}_{p(\boldsymbol{\gamma}|\boldsymbol{\phi})} \log p(\mathbf{y}_i \mid T_{\boldsymbol{\gamma}}(\mathbf{x}_i), \boldsymbol{\theta})$. Furthermore, $\Delta > 0$ when $p(\mathbf{y}_i \mid T_{\boldsymbol{\gamma}}(\mathbf{x}_i), \boldsymbol{\theta})$ varies across $\boldsymbol{\gamma}$, indicating a strictly better generalization bound for OPTIMA.*

The proofs for Theorem 4.4 and Theorem 4.5 are detailed in Appendix B.2 and Appendix B.3. This theorem provides several key insights:

**Corollary 4.6** (Marginalization Advantage). *For a fixed $\boldsymbol{\phi}$, the term $\Delta_{\boldsymbol{\phi}}(\mathbf{x}_i, \mathbf{y}_i) = \log \mathbb{E}_{p(\boldsymbol{\gamma}|\boldsymbol{\phi})} p(\mathbf{y}_i \mid T_{\boldsymbol{\gamma}}(\mathbf{x}_i), \boldsymbol{\theta}) - \mathbb{E}_{p(\boldsymbol{\gamma}|\boldsymbol{\phi})} \log p(\mathbf{y}_i \mid T_{\boldsymbol{\gamma}}(\mathbf{x}_i), \boldsymbol{\theta})$ quantifies the advantage of proper marginalization over naïve DA. By Jensen's inequality, $\Delta_{\boldsymbol{\phi}}(\mathbf{x}_i, \mathbf{y}_i) \geq 0$, with equality only when $p(\mathbf{y}_i \mid T_{\boldsymbol{\gamma}}(\mathbf{x}_i), \boldsymbol{\theta})$ is constant across all $\boldsymbol{\gamma}$ in the support of $p(\boldsymbol{\gamma}|\boldsymbol{\phi})$.*

**Corollary 4.7** (Augmentation-Aware Prior). *The PAC-Bayes bound is smallest when the prior $p(\boldsymbol{\theta}, \boldsymbol{\phi})$ reflects the invariances induced by the augmentation family. Priors that favor parameters satisfying $p(\mathbf{y} \mid T_{\boldsymbol{\gamma}}(\mathbf{x}), \boldsymbol{\theta}) \approx p(\mathbf{y} \mid \mathbf{x}, \boldsymbol{\theta})$ lead to smaller KL terms and tighter bounds.*

These results demonstrate that OPTIMA provides better generalization guarantees than naïve DA and suggests principles for designing priors that complement the DA strategy.

## 4.3 INVARIANCE ANALYSIS

We now analyze how OPTIMA promotes invariance to transformations, extending beyond first-order (Jacobian-based) analysis to include higher-order effects. This analysis reveals how the model's sensitivity to input transformations is regularized, encouraging robustness and generalization.

**Theorem 4.8** (Higher-Order Invariance). *Let $f_{\boldsymbol{\theta}}$ be a twice-differentiable function parameterized by $\boldsymbol{\theta}$, with its Hessian bounded such that $\|\nabla^2 f_{\boldsymbol{\theta}}\| \leq H$. For input transformations $T_{\boldsymbol{\gamma}}(\mathbf{x}) = x + \delta(\boldsymbol{\gamma})$, where $\delta(\boldsymbol{\gamma})$ is a perturbation with zero mean, $\mathbb{E}_{p(\boldsymbol{\gamma}|\boldsymbol{\phi})}[\delta] = 0$, and covariance $\mathbb{E}_{p(\boldsymbol{\gamma}|\boldsymbol{\phi})}[\delta \delta^{\top}] = \Sigma_{\boldsymbol{\phi}}$, the expected squared difference in the model's output under these transformations is:*

$$\mathbb{E}_{p(\boldsymbol{\gamma}|\boldsymbol{\phi})} \left[ \|f_{\boldsymbol{\theta}}(T_{\boldsymbol{\gamma}}(\mathbf{x})) - f_{\boldsymbol{\theta}}(\mathbf{x})\|^2 \right] = \mathrm{Tr}\left( J_f(\mathbf{x})^{\top} J_f(\mathbf{x}) \Sigma_{\boldsymbol{\phi}} \right)$$

$$+ \frac{1}{4} \mathbb{E}_{p(\boldsymbol{\gamma}|\boldsymbol{\phi})} \left[ \delta^{\top} \nabla^2 f_{\boldsymbol{\theta}}(\mathbf{x})^{\top} \nabla^2 f_{\boldsymbol{\theta}}(\mathbf{x}) \delta \right] + \mathcal{O}(\|\delta\|^3), \tag{11}$$

*where $J_f(\mathbf{x})$ is the Jacobian of $f_{\boldsymbol{\theta}}$ at input $x$, $\nabla^2 f_{\boldsymbol{\theta}}(\mathbf{x})$ is the Hessian of $f_{\boldsymbol{\theta}}$ at $x$, and $\mathcal{O}(\|\delta\|^3)$ represents higher-order terms that become negligible for small perturbations.*

**Corollary 4.9** (Input-Space Regularization). *The second-order term in Theorem 4.8 acts as a regularizer, penalizing high curvature in the model's output with respect to the input. This encourages a smoother response surface, promoting robustness to transformations and potentially enhancing generalization by reducing sensitivity to irrelevant input variations.*

**Corollary 4.10** (Optimal Transformation Covariance). *The optimal covariance structure $\Sigma_\phi$ for the augmentation distribution depends on the geometry of the model's response surface. Specifically, $\Sigma_\phi$ should allocate more variance in directions where the model is approximately invariant (small eigenvalues of $J_f(\mathbf{x})^\top J_f(\mathbf{x})$) and less variance in directions of high sensitivity.*

*Remark* 4.11. In our framework, $\phi$ is inferred via $q(\phi)$ by maximizing the augmented ELBO, allowing the DA distribution to adapt to the data and further enhance model robustness.

It provides practical guidance for designing augmentation distributions that align with the model's natural invariances, enhancing robustness and generalization. The proof can be found in Appendix B.4.

## 4.4 MARGINALIZATION VS. HEURISTIC AUGMENTATION

We now quantify the difference between our marginalization approach and naïve DA, focusing on the impact on posterior uncertainty. The next theorem assumes local Gaussianity of the posterior with full-rank covariance, which might not be the case in practice for over-parameterized models; however, we believe that the theoretical development gives some useful insights into the behavior of OPTIMA compared to naïve DA, and we will attempt a more general proof in the future.

**Theorem 4.12** (Posterior Shrinkage under Naïve Augmentation). *Let $p_{true}(\boldsymbol{\theta} \mid \mathcal{D})$ be the posterior under our marginalization approach and $p_{naïve}(\boldsymbol{\theta} \mid \mathcal{D})$ be the posterior under naïve DA with $K$ augmentations per data point. Under regularity conditions and assuming a locally Gaussian approximation around the MAP estimate $\hat{\boldsymbol{\theta}}$: $\Sigma_{naïve} \approx \frac{1}{K}\Sigma_{true}$, where $\Sigma_{naïve}$ and $\Sigma_{true}$ are full-rank covariance matrices of $p_{naïve}(\boldsymbol{\theta} \mid \mathcal{D})$ and $p_{true}(\boldsymbol{\theta} \mid \mathcal{D})$, respectively.*

The proof is in Appendix B.5. This result has significant implications for uncertainty quantification:

**Corollary 4.13** (Uncertainty Propagation). *Predictive uncertainty is underestimated by a factor of approximately $\sqrt{K}$ under naïve augmentation, leading to overconfident predictions, particularly for out-of-distribution inputs.*

These results provide a quantitative characterization of the benefits of proper marginalization over naïve augmentation, particularly for uncertainty quantification and calibration.

## 4.5 EMPIRICAL BAYES PERSPECTIVE

Finally, we analyze OPTIMA from an empirical Bayes perspective (Robbins, 1992; Efron, 2012), showing how it naturally leads to optimal augmentation strategies.

**Theorem 4.14** (Empirical Bayes Optimality via Augmented ELBO). *The augmented $\text{ELBO}_{aug}(q_{\boldsymbol{\theta}}, q_{\boldsymbol{\phi}})$ (see Eq. 7) holds when $q(\boldsymbol{\theta}) = p(\boldsymbol{\theta} \mid \mathcal{D})$ and $q(\boldsymbol{\phi}) = p(\boldsymbol{\phi} \mid \mathcal{D})$. Consequently, maximizing $\text{ELBO}_{aug}(q_{\boldsymbol{\theta}}, q_{\boldsymbol{\phi}})$ with respect to both $q(\boldsymbol{\theta})$ and $q(\boldsymbol{\phi})$ approximates the posterior distributions $p(\boldsymbol{\theta} \mid \mathcal{D})$ and $p(\boldsymbol{\phi} \mid \mathcal{D})$, with the mode or mean of $q(\boldsymbol{\phi})$ serving as a point estimate analogous to an Empirical Bayes solution, regularized by the prior $p(\boldsymbol{\phi})$.*

**Corollary 4.15** (Data-Driven Augmentation). *The optimization of $q(\boldsymbol{\phi})$ via the augmented ELBO results in an augmentation distribution $p(\boldsymbol{\gamma} \mid \boldsymbol{\phi})$ that is specifically tailored to the observed data $\mathcal{D}$, with $\boldsymbol{\phi} \sim q(\boldsymbol{\phi})$. This process effectively selects DA parameters enhancing the ability of the model to explain the data, implicitly performs model selection over the space of augmentation strategies.*

**Corollary 4.16** (Convergence of Joint Optimization). *Under mild regularity conditions (e.g., continuity and boundedness of the likelihood and prior), the alternating optimization of the variational distributions $q(\boldsymbol{\theta})$ and $q(\boldsymbol{\phi})$ converges to a local optimum of the marginal likelihood $p(\mathcal{D})$. This ensures that the learned augmentation distribution $p(\boldsymbol{\gamma} \mid \boldsymbol{\phi})$ is both data-consistent and aligned with the model's posterior distribution.*

These results establish OPTIMA as a principled, data-driven method for learning optimal augmentation strategies within a Bayesian framework. The proof is detailed in Appendix B.6.

### 4.6 INFORMATION-THEORETIC PERSPECTIVE

We now provide an information-theoretic analysis of OPTIMA, offering additional insights into the role of DA in Bayesian inference.

**Theorem 4.17** (Information Gain from Augmentation). *The expected information gain from* DA, *measured as the reduction in posterior entropy, is:*

$$\Delta H = H[p(\boldsymbol{\theta} \,|\, \mathcal{D}_{\text{noaug}})] - H[p(\boldsymbol{\theta} \,|\, \mathcal{D})] \approx \frac{1}{2} \log \det(I + H_{\text{noaug}}^{-1} H_{\text{aug}}), \tag{12}$$

*where* $H_{\text{noaug}}$ *and* $H_{\text{aug}}$ *are the Hessians of the negative log-likelihood without and with* DA, *respectively, and* $p(\boldsymbol{\theta} \,|\, \mathcal{D})$ *uses the marginalized likelihood.*

The proof is in Appendix B.7. This information-theoretic perspective provides additional insights:

**Corollary 4.18** (Optimal Information Gain). *The* DA *distribution that maximizes information gain while maintaining a fixed KL divergence from a reference distribution aligns with the eigenvectors of the Fisher information matrix, with variance inversely proportional to the eigenvalues.*

**Corollary 4.19** (Connection to Information Bottleneck). OPTIMA *can be viewed as implementing an information bottleneck, where the* DA *distribution* $p(\boldsymbol{\gamma}|\boldsymbol{\phi})$ *is optimized to maximize the mutual information between the augmented inputs and the targets, while minimizing the mutual information between the original and augmented inputs.*

These information-theoretic results provide a complementary perspective on OPTIMA, connecting it to principles of optimal experimental design and information bottleneck theory.

## 5 EXPERIMENTS

### 5.1 SYNTHETIC REGRESSION EXAMPLE

We begin with a toy regression problem by generating 50 training and 1000 test points from $y = \sin(2x) + 0.5\cos(3x) + \varepsilon + \epsilon \sin(x)$ with $\varepsilon \sim \mathcal{N}(0, 0.2^2)$ and $\epsilon \sim \mathcal{N}(0, 0.15^2)$. We report results in Fig. 2. In the competing approaches, **Fixed Aug** augments data by adding Gaussian noise with fixed standard deviation $\sigma = 0.1$. In **Naïve Aug**, for each training example, we average the loss over $K = 5$ independent augmentations with $\sigma = 0.1$. In OPTIMA, the augmentation shift $\gamma \sim \mathcal{N}(\mu, \sigma^2)$ has learnable parameters and it has a prior $\mathcal{N}(0, 0.2^2)$.

Although **No Aug** attains a lower training error, its test error is significantly higher due to overfitting. Conversely, **Fixed Aug** and **Naïve Aug** achieve better test performance than no augmentation, indicating that input perturbations help regularize the model. Our OPTIMA achieves competitive test MSE. The learned augmentation distribution widens over training taking $\sigma$ from $0.10$ to about $0.18$, implying that a broader range of translational perturbations is optimal for this dataset. This dynamic adaption shows the benefit of OPTIMA's ability to learn the augmentation distribution, and it is theoretically justified in Corollary 4.2 and Corollary 4.15, which state that OPTIMA tailors the augmentation distribution to the observed data. For additional ablations with different intensities on image classification dataset CIFAR10, see Appendix F.2.

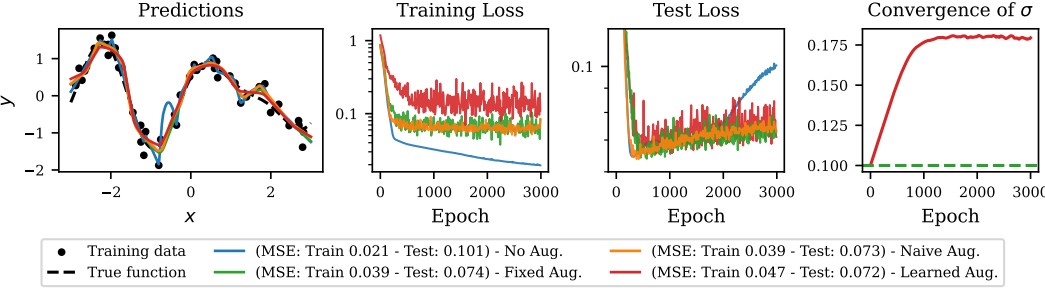

Figure 2: Synthetic regression: *(Left)* Test predictions compared to the ground-truth function. *(Right)* Convergence traces for OPTIMA; green dashed line denotes the fixed $\sigma = 0.1$ used in **Fixed Aug**.

## 5.2 IMAGENET AND IMAGENET-C

We next evaluate the robustness of OPTIMA on IMA-
GENET (Deng et al., 2009) and IMAGENET-C, an out-
of-distribution (OOD) dataset (Hendrycks & Dietterich,
2019) using a Bayesian ResNet-18 (He et al., 2016), where
the final layer is replaced with a `BayesianLinear`
module. This partially stochastic design—treating only
the final layer in a Bayesian manner—is a common and

Table 1: IMAGENET and IMAGENET-C
with non-Bayesian ResNet-50.

| Method | Acc (%) Clean | Acc (%) Corrupted |
|---|---|---|
| Mixup | 76.1 | 40.1 |
| OPTIMA Mixup | **76.8** | **41.6** |

efficient strategy in Bayesian deep learning (Harrison et al., 2024). As noted by Sharma et al. (2023),
full network stochasticity is often unnecessary; introducing stochasticity in the final layer can be
sufficient to capture predictive uncertainty, especially with strong deterministic feature extractors
(Kristiadi et al., 2020). This allows the model to represent uncertainty in class probabilities while
leveraging the pretrained backbone. With OPTIMA, we optimize augmentation parameters for
Mixup (Zhang et al., 2018), CutMix (Yun et al., 2019), and AugMix (Hendrycks et al., 2020). Im-
portantly, our approach is general and can also be applied to standard (non-Bayesian) networks.
To illustrate this, we further evaluate OPTIMA on ResNet-50 without Bayesian treatment of its
parameters. Implementation details are provided in Appendix E.

Table 2 summarizes the results, confirming that OPTIMA obtains better calibration and robustness
for both ID and OOD. Regarding the non-Bayesian NNs, our framework also allows us to have better
accuracy on the clean and corrupted data (see Table 1), further demonstrating that OPTIMA captures
variations in the data better than fixed augmentations. More results can be found in Appendix G.5.
These results supports our theoretical analyses – e.g, Theorem 4.5 (improved generalization on test
and OOD data) and Theorem 4.12 (enhanced calibration and uncertainty quantification).

Table 2: IMAGENET and IMAGENET-C with pretrained Bayesian ResNet-50 (last layer) after 10
epochs ("C"-corrupted, "m" - mean)

| Method | Test Acc (Clean Data) (%) | ECE ($\downarrow$) | mCE ($\downarrow$) (unnormalized) (C) (%) | mECE ($\downarrow$) (C) | mOOD-AUROC (C) |
|---|---|---|---|---|---|
| Fixed Mixup | 75.39 | 0.043 | 61.69 | 0.062 | 0.820 |
| OPTIMA Mixup | 74.97 | **0.031** | **61.65** | **0.045** | **0.822** |
| Fixed Cutmix | 74.17 | 0.036 | 63.28 | 0.059 | 0.819 |
| OPTIMA Cutmix | **74.34** | **0.034** | 63.60 | **0.058** | **0.820** |
| Fixed Augmix | 74.71 | 0.084 | 61.45 | 0.156 | 0.790 |
| OPTIMA Augmix | **75.33** | **0.083** | **60.68** | **0.149** | **0.793** |

## 5.3 COMPUTATIONAL EFFICIENCY AND COMPARISON WITH BAYESIAN OPTIMIZATION

Our method introduces almost no additional computational cost compared to traditional data augmen-
tation. The difference lies in our adaptive augmentation strategies, which evolve over iterations rather
than remaining fixed. OPTIMA employs Monte Carlo estimates with a small sample size like one
per data point per iteration and uses the reparameterization trick for efficient, low-variance gradient
estimation. To highlight its efficiency, we compare against Bayesian Optimization (BO), a strong
baseline for augmentation tuning. BO requires costly black-box optimization with many full training
runs per hyperparameter setting, whereas OPTIMA's tractable ELBO jointly optimizes augmentation
and model parameters within the same training loop—removing the need for separate validation runs.

We evaluate on CIFAR10 (Krizhevsky & Hinton, 2009) using a pretrained Bayesian ResNet-18
(Bayesian last layer) to optimize augmentation parameters (mean and variance). BO is run for
25 trials of 15 epochs, followed by 50 epochs of training with the optimized parameters, while
OPTIMA is trained directly for 50 epochs. For augmentation, we use Mixup and learn the parameter
$\alpha$. We also assess performance on CIFAR10-C (Hendrycks & Dietterich, 2019) as OOD data. As
shown in Table 3, OPTIMA achieves higher test accuracy on clean data (with a slight calibration
trade-off) and substantially better accuracy, ECE, and AUROC on OOD data, all in far less time than
BO—demonstrating improved calibration and robustness at much lower cost.

Table 3: Comparison between Bayesian optimization and OPTIMA on CIFAR10

| Method | Test Acc (%) | ECE (%) | mAccuracy (C) | mECE (C) | OOD AUROC | Time |
|---|---|---|---|---|---|---|
| Bayesian Optimization | 93.43 | 0.010 | 72.44 | 0.127 | 0.652 | $\sim 4 \times T$ |
| OPTIMA | **95.03** | 0.047 | **78.52** | **0.076** | **0.680** | $T$ |

## 5.4 OPTIMA ON DISCRETE NLP AUGMENTATIONS: SST-5 CASE STUDY

To show that OPTIMA is not restricted to continuous or geometric transformations used in computer vision, we additionally evaluate it on a NLP classification task where augmentations are inherently *discrete*. We use the SST-5 benchmark (Socher et al. 2013), a fine-grained 5-class sentiment dataset, and fine-tune a DistilBERT model (Sanh et al. 2019) for 5 epochs on the full training split.

**Discrete augmentation family.**  We consider *token dropout*, a stochastic masking transformation used in NLP regularization. For input $x = (x_1, \ldots, x_L)$ with dropout rate $p_{\mathrm{drop}}$, we sample $\gamma_t \sim \mathrm{Bernoulli}(1 - p_{\mathrm{drop}})$ and replace $x_t$ with [MASK] if $\gamma_t = 0$, yielding a discrete latent variable $\gamma$. Although non-differentiable, OPTIMA optimizes $p_{\mathrm{drop}}$ via a score-function (REINFORCE) gradient, consistent with Section 3.

**Experimental setup.**  We evaluate OPTIMA on discrete token-dropout augmentation. The augmentation uses a dropout probability $p_{\mathrm{drop}} \in [0, p_{\max}]$ parameterized as $p_{\mathrm{drop}} = p_{\max} \sigma(s)$, where $s$ is a learnable scalar. To encode prior preferences for dropout, we place a Gaussian prior directly on $p_{\mathrm{drop}}$, and OPTIMA jointly learns $s$ and the model parameters via our ELBO. We compare OPTIMA against: (i) *No Aug*; (ii) *Fixed Aug*, which uses the same initial dropout as OPTIMA; (iii) *Fixed Aug (Matched)*, where $p_{\mathrm{drop}}$ is set equal to the value learned by OPTIMA; and (iv) *BO-Fixed*, which selects $p_{\mathrm{drop}}$ through a validation-based hyperparameter search. More details are provided in Appendix H.

Table 4: SST-5 results for discrete token-dropout augmentation averaged over 5 different seeds.

| Method | Accuracy | NLL | ECE |
|---|---|---|---|
| No Aug | $0.516 \pm 0.003$ | $1.240 \pm 0.010$ | $0.190 \pm 0.004$ |
| Fixed $p_{\mathrm{drop}} = 0.04$ | $0.522 \pm 0.003$ | $1.180 \pm 0.006$ | $0.154 \pm 0.006$ |
| Fixed $p_{\mathrm{drop}} = 0.0625$ | $0.516 \pm 0.006$ | $1.162 \pm 0.007$ | $0.143 \pm 0.007$ |
| OPTIMA with $\mu = 0.1$ ($p_{\mathrm{learned}} = 0.0625$) | $\mathbf{0.524 \pm 0.003}$ | $\mathbf{1.161 \pm 0.007}$ | $\mathbf{0.142 \pm 0.006}$ |
| BO-Fixed $p_{\mathrm{drop}} = 0.3$ | $0.521 \pm 0.004$ | $1.086 \pm 0.006$ | $\mathbf{0.043 \pm 0.004}$ |
| OPTIMA with $\mu = 0.3$ ($p_{\mathrm{learned}} = 0.3$) | $\mathbf{0.524 \pm 0.004}$ | $\mathbf{1.086 \pm 0.005}$ | $\mathbf{0.046 \pm 0.002}$ |

**Results.**  Table 4 shows that accuracy differences on SST-5 are small, but OPTIMA consistently achieves lower *NLL* and improved calibration over fixed-augmentation baselines. Notably, it matches the BO-tuned baseline—despite BO requiring a full hyperparameter search across multiple runs (approximately $8\times$ more compute)—while OPTIMA learns $p_{\mathrm{drop}}$ in a single training run. This indicates that the gains stem from marginal likelihood optimization rather than dropout tuning. These findings confirm that OPTIMA extends naturally to discrete augmentation spaces and that its theoretical advantages (Sections 4.2–4.3) generalize beyond vision tasks.

## 6 DISCUSSION AND CONCLUSION

We presented a theoretical and methodological framework for optimizing DA taking inspiration from Bayesian principles, which allow us to cast this problem as model selection. We derived a variational objective to learn optimal DA strategies from data in a practical way. We also provided extensive theoretical insights on the advantages of our proposed data-driven approach to DA compared to alternatives, revealing improved generalization through PAC-Bayes bounds, enhanced invariance via higher-order regularization, and better calibration through marginalization. Empirical results confirm these theoretical benefits, showing consistent improvements in calibration and predictive performance across various tasks. We believe that OPTIMA is a key step toward robust and well-calibrated models capable of assisting decision-making in applications where this is of critical importance.

**Limitations and future work.**  While OPTIMA offers significant advantages, it has some limitations that suggest directions for future work. Although our main experiments focus on computer vision, OPTIMA itself is not tied to continuous or geometric transformations. Our additional evaluation on a natural language task (SST-5) demonstrates that OPTIMA can also handle discrete, non-geometric transformations by optimizing a latent augmentation distribution in text space. Nevertheless, a broader exploration of more expressive or compositional transformations in NLP, time series, or multimodal settings remains an important next step. In addition, our theoretical analysis could be strengthened by developing tighter PAC-Bayes bounds and more refined characterizations of the benefits introduced by Bayesian marginalization over augmentation parameters.

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

# Appendix

## TABLE OF CONTENTS

## A   DERIVATION OF THE AUGMENTED EVIDENCE LOWER BOUND

For variational inference, we introduce a variational distribution $q(\boldsymbol{\theta}, \boldsymbol{\phi}) = q(\boldsymbol{\theta})q(\boldsymbol{\phi})$ to approximate the posterior $p(\boldsymbol{\theta}, \boldsymbol{\phi} \,|\, \mathcal{D})$. The standard ELBO is a lower bound on the log marginal likelihood

$\log p(\mathcal{D}) = \log \iiint p(\mathcal{D}, \boldsymbol{\theta}, \boldsymbol{\phi}, \boldsymbol{\gamma}) \, d\boldsymbol{\theta} \, d\boldsymbol{\phi} \, d\boldsymbol{\gamma}$. Using Jensen's inequality with $q(\boldsymbol{\theta}, \boldsymbol{\phi})$, we have:

$$\log p(\mathcal{D}) \geq \underbrace{\mathbb{E}_{q(\boldsymbol{\theta}, \boldsymbol{\phi})} \left[ \log \frac{p(\mathcal{D}, \boldsymbol{\theta}, \boldsymbol{\phi})}{q(\boldsymbol{\theta}, \boldsymbol{\phi})} \right]}_{\text{ELBO}(q)}, \quad \text{where } p(\mathcal{D}, \boldsymbol{\theta}, \boldsymbol{\phi}) = \int p(\mathcal{D}, \boldsymbol{\theta}, \boldsymbol{\phi}, \boldsymbol{\gamma}) \, d\boldsymbol{\gamma}. \quad (13)$$

Applying Jensen's inequality further to the log of the likelihood term, $p(\mathcal{D} \,|\, \boldsymbol{\theta}, \boldsymbol{\phi})$:

$$\log p(\mathcal{D} \,|\, \boldsymbol{\theta}, \boldsymbol{\phi}) = \log \mathbb{E}_{p(\boldsymbol{\gamma} \,|\, \boldsymbol{\phi})} \left[ \prod_{i=1}^{N} p(\mathbf{y}_i \,|\, T_{\boldsymbol{\gamma}}(\mathbf{x}_i), \boldsymbol{\theta}) \right] \geq \mathbb{E}_{p(\boldsymbol{\gamma} \,|\, \boldsymbol{\phi})} \left[ \sum_{i=1}^{N} \log p(\mathbf{y}_i \,|\, T_{\boldsymbol{\gamma}}(\mathbf{x}_i), \boldsymbol{\theta}) \right]. \quad (14)$$

Substituting this into the ELBO in Eq. 13, with $p(\mathcal{D}, \boldsymbol{\theta}, \boldsymbol{\phi}) = p(\mathcal{D} \,|\, \boldsymbol{\theta}, \boldsymbol{\phi})p(\boldsymbol{\theta})p(\boldsymbol{\phi})$, we can obtain the augmented ELBO as follows:

$$\text{ELBO}(q) \geq \mathbb{E}_{q(\boldsymbol{\theta}, \boldsymbol{\phi})} \left[ \mathbb{E}_{p(\boldsymbol{\gamma} \,|\, \boldsymbol{\phi})} \left[ \sum_{i=1}^{N} \log p(\mathbf{y}_i \,|\, T_{\boldsymbol{\gamma}}(\mathbf{x}_i), \boldsymbol{\theta}) \right] + \log \frac{p(\boldsymbol{\theta})p(\boldsymbol{\phi})}{q(\boldsymbol{\theta})q(\boldsymbol{\phi})} \right] \quad (15)$$

$$= \underbrace{\mathbb{E}_{q(\boldsymbol{\theta})} \mathbb{E}_{q(\boldsymbol{\phi})} \mathbb{E}_{p(\boldsymbol{\gamma} \,|\, \boldsymbol{\phi})} \left[ \sum_{i=1}^{N} \log p(\mathbf{y}_i \,|\, T_{\boldsymbol{\gamma}}(\mathbf{x}_i), \boldsymbol{\theta}) \right]}_{\text{data fit}} - \underbrace{\text{KL}(q(\boldsymbol{\theta}) \| p(\boldsymbol{\theta}))}_{\text{parameter prior}} - \underbrace{\text{KL}(q(\boldsymbol{\phi}) \| p(\boldsymbol{\phi}))}_{\text{augmentation prior}}. \quad (16)$$

The augmented ELBO consists of three terms: a data-fitting term that averages over the variational distributions $q(\boldsymbol{\theta})$, $q(\boldsymbol{\phi})$, and the augmentation distribution $p(\boldsymbol{\gamma} \,|\, \boldsymbol{\phi})$; a regularization term $\text{KL}(q(\boldsymbol{\theta}) \| p(\boldsymbol{\theta}))$ that penalizes divergence from the prior over model parameters; and another regularization term $\text{KL}(q(\boldsymbol{\phi}) \| p(\boldsymbol{\phi}))$ that aligns the augmentation parameters with their prior.

## B  DETAILED PROOFS

This section provides expanded proofs for the theoretical results presented in the main paper.

### B.1  PROOF OF PROPOSITION 4.1 (JENSEN GAP BOUND)

*Proof.* Let $f(\boldsymbol{\gamma}) = \log p(\mathbf{y} \,|\, T_{\boldsymbol{\gamma}}(\mathbf{x}), \boldsymbol{\theta})$. For sub-Gaussian $\boldsymbol{\gamma}$ with mean $\mu = \mathbb{E}[\boldsymbol{\gamma}]$ and variance proxy $\sigma^2$, we use standard moment generating function bounds:

$$\log \mathbb{E}_{\boldsymbol{\gamma}}[e^{f(\boldsymbol{\gamma})}] = \log \mathbb{E}_{\boldsymbol{\gamma}} \left[ e^{f(\mu) + (f(\boldsymbol{\gamma}) - f(\mu))} \right] \quad (17)$$

$$= f(\mu) + \log \mathbb{E}_{\boldsymbol{\gamma}} \left[ e^{f(\boldsymbol{\gamma}) - f(\mu)} \right]. \quad (18)$$

Since $f$ is $L$-Lipschitz, we have $|f(\boldsymbol{\gamma}) - f(\mu)| \leq L\|\boldsymbol{\gamma} - \mu\|$. Using the sub-Gaussian property:

$$\mathbb{E}_{\boldsymbol{\gamma}}[e^{f(\boldsymbol{\gamma}) - f(\mu)}] \leq \mathbb{E}_{\boldsymbol{\gamma}}[e^{L\|\boldsymbol{\gamma} - \mu\|}] \leq e^{\frac{L^2 \sigma^2}{2}}. \quad (19)$$

Therefore:

$$\log \mathbb{E}_{\boldsymbol{\gamma}}[e^{f(\boldsymbol{\gamma})}] \leq f(\mu) + \frac{L^2 \sigma^2}{2}. \quad (20)$$

Since $\mathbb{E}_{\boldsymbol{\gamma}}[f(\boldsymbol{\gamma})] = f(\mu)$ when $\mathbb{E}_{\boldsymbol{\gamma}}[\boldsymbol{\gamma} - \mu] = 0$, the gap is:

$$\text{Gap} = \log \mathbb{E}_{\boldsymbol{\gamma}}[e^{f(\boldsymbol{\gamma})}] - \mathbb{E}_{\boldsymbol{\gamma}}[f(\boldsymbol{\gamma})] \quad (21)$$

$$\leq \frac{L^2 \sigma^2}{2}. \quad (22)$$

For tightness, when $f(\boldsymbol{\gamma})$ is approximately linear in the high-probability region of $p(\boldsymbol{\gamma}|\boldsymbol{\phi})$, the bound approaches equality. $\square$

## B.2 PROOF OF THEOREM 4.4 (PAC-BAYES UNDER AUGMENTATION)

*Proof.* Define the loss function as:

$$\ell(\boldsymbol{\theta}, \boldsymbol{\phi}, (x, y)) = -\log \mathbb{E}_{p(\boldsymbol{\gamma}|\boldsymbol{\phi})}[p(y \mid T_{\boldsymbol{\gamma}}(x), \boldsymbol{\theta})]. \tag{23}$$

The empirical risk is:

$$\hat{R}(\boldsymbol{\theta}, \boldsymbol{\phi}) = \frac{1}{N} \sum_{i=1}^{N} \ell(\boldsymbol{\theta}, \boldsymbol{\phi}, (\mathbf{x}_i, \mathbf{y}_i)). \tag{24}$$

Since $\{(\mathbf{x}_i, \mathbf{y}_i)\}_{i=1}^{N}$ are i.i.d., and the expectation over $\boldsymbol{\gamma} \sim p(\boldsymbol{\gamma} \mid \boldsymbol{\phi})$ is computed independently for each sample, the terms $\ell(\boldsymbol{\theta}, \boldsymbol{\phi}, (\mathbf{x}_i, \mathbf{y}_i))$ are independent for fixed $\boldsymbol{\theta}, \boldsymbol{\phi}$. Applying the standard PAC-Bayes theorem over the joint space $(\boldsymbol{\theta}, \boldsymbol{\phi})$ (McAllester, 1999; Catoni, 2007; Alquier, 2024):

$$\mathbb{E}_{q(\boldsymbol{\theta}, \boldsymbol{\phi})}[R(\boldsymbol{\theta}, \boldsymbol{\phi})] \leq \mathbb{E}_{q(\boldsymbol{\theta}, \boldsymbol{\phi})}[\hat{R}(\boldsymbol{\theta}, \boldsymbol{\phi})] + \sqrt{\frac{\text{KL}(q(\boldsymbol{\theta}, \boldsymbol{\phi}) \| p(\boldsymbol{\theta}, \boldsymbol{\phi})) + \log \frac{2\sqrt{N}}{\delta}}{2N}}. \tag{25}$$

This completes the proof. $\qquad\square$

## B.3 PROOF OF THEOREM 4.5 (GENERALIZATION ADVANTANGE OF BAYESIAN-OPTIMIZED AUGMENTATION)

*Proof.* **Step 1: PAC-Bayes Bounds**

For OPTIMA, the PAC-Bayes bound is:

$$\mathbb{E}_{q(\boldsymbol{\theta}, \boldsymbol{\phi})}[R(\boldsymbol{\theta}, \boldsymbol{\phi})] \leq \mathbb{E}_{q(\boldsymbol{\theta}, \boldsymbol{\phi})}[\hat{R}(\boldsymbol{\theta}, \boldsymbol{\phi})] + \sqrt{\frac{\text{KL}(q(\boldsymbol{\theta}, \boldsymbol{\phi}) \| p(\boldsymbol{\theta}, \boldsymbol{\phi})) + \log \frac{2\sqrt{N}}{\delta}}{2N}}. \tag{26}$$

For naïve augmentation, the bound is:

$$\mathbb{E}_{q(\boldsymbol{\theta})}[R(\boldsymbol{\theta})] \leq \mathbb{E}_{q(\boldsymbol{\theta})}[\hat{R}_{\text{naïve}}(\boldsymbol{\theta})] + \sqrt{\frac{\text{KL}(q(\boldsymbol{\theta}) \| p(\boldsymbol{\theta})) + \log \frac{2\sqrt{N}}{\delta}}{2N}}. \tag{27}$$

**Step 2: Relationship Between Empirical Risks**

By Jensen's inequality, for each data point $(\mathbf{x}_i, \mathbf{y}_i)$:

$$\log \mathbb{E}_{p(\boldsymbol{\gamma}|\boldsymbol{\phi})} p(\mathbf{y}_i \mid T_{\boldsymbol{\gamma}}(\mathbf{x}_i), \boldsymbol{\theta}) \geq \mathbb{E}_{p(\boldsymbol{\gamma}|\boldsymbol{\phi})} \log p(\mathbf{y}_i \mid T_{\boldsymbol{\gamma}}(\mathbf{x}_i), \boldsymbol{\theta}). \tag{28}$$

Thus,

$$-\log \mathbb{E}_{p(\boldsymbol{\gamma}|\boldsymbol{\phi})} p(\mathbf{y}_i \mid T_{\boldsymbol{\gamma}}(\mathbf{x}_i), \boldsymbol{\theta}) \leq -\mathbb{E}_{p(\boldsymbol{\gamma}|\boldsymbol{\phi})} \log p(\mathbf{y}_i \mid T_{\boldsymbol{\gamma}}(\mathbf{x}_i), \boldsymbol{\theta}). \tag{29}$$

For large $K$, the naïve empirical risk approximates:

$$\hat{R}_{\text{naïve}}(\boldsymbol{\theta}) \approx -\frac{1}{N} \sum_{i=1}^{N} \mathbb{E}_{p(\boldsymbol{\gamma}|\boldsymbol{\phi})} \log p(\mathbf{y}_i \mid T_{\boldsymbol{\gamma}}(\mathbf{x}_i), \boldsymbol{\theta}). \tag{30}$$

Therefore,

$$\hat{R}(\boldsymbol{\theta}, \boldsymbol{\phi}) = -\frac{1}{N} \sum_{i=1}^{N} \log \mathbb{E}_{p(\boldsymbol{\gamma}|\boldsymbol{\phi})} p(\mathbf{y}_i \mid T_{\boldsymbol{\gamma}}(\mathbf{x}_i), \boldsymbol{\theta}) \le \hat{R}_{\text{naïve}}(\boldsymbol{\theta}), \tag{31}$$

with equality only if $\Delta_{\boldsymbol{\phi}}(\mathbf{x}_i, \mathbf{y}_i) = 0$ for all $i$, i.e., when $p(\mathbf{y}_i \mid T_{\boldsymbol{\gamma}}(\mathbf{x}_i), \boldsymbol{\theta})$ is constant across $\boldsymbol{\gamma}$.

**Step 3: Bound Comparison**

Assuming $\text{KL}(q(\boldsymbol{\theta}, \boldsymbol{\phi}) \| p(\boldsymbol{\theta}, \boldsymbol{\phi})) \approx \text{KL}(q(\boldsymbol{\theta}) \| p(\boldsymbol{\theta}))$, the difference in bounds is driven by the empirical risks:

$$\mathbb{E}_{q(\boldsymbol{\theta}, \boldsymbol{\phi})}[R(\boldsymbol{\theta}, \boldsymbol{\phi})] \le \mathbb{E}_{q(\boldsymbol{\theta}, \boldsymbol{\phi})}[\hat{R}(\boldsymbol{\theta}, \boldsymbol{\phi})] + \text{complexity term}, \tag{32}$$

$$\mathbb{E}_{q(\boldsymbol{\theta})}[R(\boldsymbol{\theta})] \le \mathbb{E}_{q(\boldsymbol{\theta})}[\hat{R}_{\text{naïve}}(\boldsymbol{\theta})] + \text{complexity term}. \tag{33}$$

Since $\hat{R}(\boldsymbol{\theta}, \boldsymbol{\phi}) \le \hat{R}_{\text{naïve}}(\boldsymbol{\theta})$, and the complexity terms are similar, our bound is tighter. Specifically,

$$\mathbb{E}_{q(\boldsymbol{\theta}, \boldsymbol{\phi})}[\hat{R}(\boldsymbol{\theta}, \boldsymbol{\phi})] = \mathbb{E}_{q(\boldsymbol{\theta}, \boldsymbol{\phi})}[\hat{R}_{\text{naïve}}(\boldsymbol{\theta})] - \mathbb{E}_{q(\boldsymbol{\theta}, \boldsymbol{\phi})}\left[\frac{1}{N} \sum_{i=1}^{N} \Delta_{\boldsymbol{\phi}}(\mathbf{x}_i, \mathbf{y}_i)\right], \tag{34}$$

leading to:

$$\mathbb{E}_{q(\boldsymbol{\theta}, \boldsymbol{\phi})}[R(\boldsymbol{\theta}, \boldsymbol{\phi})] \le \mathbb{E}_{q(\boldsymbol{\theta})}[\hat{R}_{\text{naïve}}(\boldsymbol{\theta})] - \Delta + \text{complexity term}, \tag{35}$$

where $\Delta = \mathbb{E}_{q(\boldsymbol{\theta}, \boldsymbol{\phi})}\left[\frac{1}{N} \sum_{i=1}^{N} \Delta_{\boldsymbol{\phi}}(\mathbf{x}_i, \mathbf{y}_i)\right] \ge 0$.

Thus, OPTIMA 's bound is lower by $\Delta$, proving better generalization. When $p(\mathbf{y}_i \mid T_{\boldsymbol{\gamma}}(\mathbf{x}_i), \boldsymbol{\theta})$ varies across $\boldsymbol{\gamma}$, $\Delta > 0$, making our bound strictly tighter. $\qquad\square$

### B.4 PROOF OF THEOREM 4.8 (HIGHER-ORDER INVARIANCE)

*Proof.* Using a second-order Taylor expansion of $f_{\boldsymbol{\theta}}$ around $x$:

$$f_{\boldsymbol{\theta}}(T_{\boldsymbol{\gamma}}(\mathbf{x})) = f_{\boldsymbol{\theta}}(\mathbf{x}) + J_f(\mathbf{x})\delta + \frac{1}{2}\delta^T \nabla^2 f_{\boldsymbol{\theta}}(\mathbf{x})\delta + \mathcal{O}(\|\delta\|^3), \tag{36}$$

$$f_{\boldsymbol{\theta}}(T_{\boldsymbol{\gamma}}(\mathbf{x})) - f_{\boldsymbol{\theta}}(\mathbf{x}) = J_f(\mathbf{x})\delta + \frac{1}{2}\delta^T \nabla^2 f_{\boldsymbol{\theta}}(\mathbf{x})\delta + \mathcal{O}(\|\delta\|^3). \tag{37}$$

Squaring this difference and taking the expectation over $p(\boldsymbol{\gamma}|\boldsymbol{\phi})$:

$$\mathbb{E}_{p(\boldsymbol{\gamma}|\boldsymbol{\phi})}\left[\|f_{\boldsymbol{\theta}}(T_{\boldsymbol{\gamma}}(\mathbf{x})) - f_{\boldsymbol{\theta}}(\mathbf{x})\|^2\right] = \mathbb{E}_{p(\boldsymbol{\gamma}|\boldsymbol{\phi})}\left[\|J_f(\mathbf{x})\delta\|^2\right] \tag{38}$$

$$+ \mathbb{E}_{p(\boldsymbol{\gamma}|\boldsymbol{\phi})}\left[\left(\frac{1}{2}\delta^T \nabla^2 f_{\boldsymbol{\theta}}(\mathbf{x})\delta\right)^2\right] \tag{39}$$

$$+ \mathbb{E}_{p(\boldsymbol{\gamma}|\boldsymbol{\phi})}\left[2\left(J_f(\mathbf{x})\delta\right)^T \left(\frac{1}{2}\delta^T \nabla^2 f_{\boldsymbol{\theta}}(\mathbf{x})\delta\right)\right] \tag{40}$$

$$+ \mathcal{O}(\|\delta\|^3). \tag{41}$$

The cross-term $\mathbb{E}_{p(\boldsymbol{\gamma}|\boldsymbol{\phi})}\left[\left(J_f(\mathbf{x})\delta\right)^T \left(\delta^T \nabla^2 f_{\boldsymbol{\theta}}(\mathbf{x})\delta\right)\right]$ involves odd powers of $\delta$, which vanish since $\mathbb{E}[\delta] = 0$. Thus:

$$\mathbb{E}_{p(\boldsymbol{\gamma}|\boldsymbol{\phi})}\left[\|f_{\boldsymbol{\theta}}(T_{\boldsymbol{\gamma}}(\mathbf{x})) - f_{\boldsymbol{\theta}}(\mathbf{x})\|^2\right] = \mathbb{E}_{p(\boldsymbol{\gamma}|\boldsymbol{\phi})}\left[\delta^T J_f(\mathbf{x})^T J_f(\mathbf{x})\delta\right] \tag{42}$$

$$+ \frac{1}{4}\mathbb{E}_{p(\boldsymbol{\gamma}|\boldsymbol{\phi})}\left[\left(\delta^T \nabla^2 f_{\boldsymbol{\theta}}(\mathbf{x})\delta\right)^2\right] + \mathcal{O}(\|\delta\|^3). \tag{43}$$

Using properties of quadratic forms for zero-mean random variables:

$$\mathbb{E}_{p(\gamma|\phi)}\left[\delta^T J_f(\mathbf{x})^T J_f(\mathbf{x})\delta\right] = \mathrm{Tr}\left(J_f(\mathbf{x})^T J_f(\mathbf{x})\Sigma_\phi\right), \tag{44}$$

$$\mathbb{E}_{p(\gamma|\phi)}\left[\left(\delta^T \nabla^2 f_{\boldsymbol{\theta}}(\mathbf{x})\delta\right)^2\right] = \mathrm{Tr}\left(\nabla^2 f_{\boldsymbol{\theta}}(\mathbf{x})^T \nabla^2 f_{\boldsymbol{\theta}}(\mathbf{x})\Sigma_\phi\right) + 2\mathrm{Tr}\left((\nabla^2 f_{\boldsymbol{\theta}}(\mathbf{x})\Sigma_\phi)^2\right). \tag{45}$$

For small perturbations, the dominant term is $\mathrm{Tr}\left(\nabla^2 f_{\boldsymbol{\theta}}(\mathbf{x})^T \nabla^2 f_{\boldsymbol{\theta}}(\mathbf{x})\Sigma_\phi\right)$, and higher moments contribute to $\mathcal{O}(\|\delta\|^3)$. Therefore:

$$\mathbb{E}_{p(\gamma|\phi)}\left[\|f_{\boldsymbol{\theta}}(T_\gamma(\mathbf{x})) - f_{\boldsymbol{\theta}}(\mathbf{x})\|^2\right] \approx \mathrm{Tr}\left(J_f(\mathbf{x})^T J_f(\mathbf{x})\Sigma_\phi\right) + \frac{1}{4}\mathrm{Tr}\left(\nabla^2 f_{\boldsymbol{\theta}}(\mathbf{x})^T \nabla^2 f_{\boldsymbol{\theta}}(\mathbf{x})\Sigma_\phi\right). \tag{46}$$

This approximation holds for small $\|\delta\|$, completing the proof. $\qquad\square$

## B.5 PROOF OF THEOREM 4.12 (POSTERIOR SHRINKAGE UNDER NAÏVE AUGMENTATION)

*Proof.* Under a locally Gaussian approximation with full-rank covariance, the posterior covariance is approximately the inverse of the Hessian of the negative log posterior at the MAP estimate. For the true posterior, marginalizing over $\phi$:

$$p(\mathcal{D}\,|\,\boldsymbol{\theta}) = \int p(\mathcal{D}\,|\,\boldsymbol{\theta},\phi)p(\phi)\,d\phi,$$

$$\Sigma_{\text{true}}^{-1} \approx -\nabla^2 \log p_{\text{true}}(\boldsymbol{\theta}\,|\,\mathcal{D})|_{\boldsymbol{\theta}=\hat{\boldsymbol{\theta}}} = -\nabla^2 \log p(\boldsymbol{\theta}) \tag{47}$$

$$-\sum_{i=1}^{N} \nabla^2 \log\left(\int \mathbb{E}_{p(\gamma|\phi)}[p(\mathbf{y}_i\,|\,T_\gamma(\mathbf{x}_i),\boldsymbol{\theta})]p(\phi)\,d\phi\right)\Big|_{\boldsymbol{\theta}=\hat{\boldsymbol{\theta}}}.$$

For the naïve posterior:

$$\Sigma_{\text{naïve}}^{-1} \approx -\nabla^2 \log p_{\text{naïve}}(\boldsymbol{\theta}\,|\,\mathcal{D})|_{\boldsymbol{\theta}=\hat{\boldsymbol{\theta}}} = -\nabla^2 \log p(\boldsymbol{\theta}) - \sum_{i=1}^{N}\sum_{k=1}^{K} \nabla^2 \log p(\mathbf{y}_i\,|\,T_{\gamma_k}(\mathbf{x}_i),\boldsymbol{\theta})|_{\boldsymbol{\theta}=\hat{\boldsymbol{\theta}}}. \tag{48}$$

Assuming $\gamma_k \sim p(\gamma\,|\,\hat{\phi})$ (e.g., using a point estimate of $\phi$), this approximates:

$$\approx -\nabla^2 \log p(\boldsymbol{\theta}) - K\sum_{i=1}^{N} \nabla^2 \log \mathbb{E}_{p(\gamma|\hat{\phi})}[p(\mathbf{y}_i\,|\,T_\gamma(\mathbf{x}_i),\boldsymbol{\theta})]|_{\boldsymbol{\theta}=\hat{\boldsymbol{\theta}}} \approx K \cdot \Sigma_{\text{true}}^{-1}. \tag{49}$$

Therefore, $\Sigma_{\text{naïve}} \approx \frac{1}{K}\Sigma_{\text{true}}$. $\qquad\square$

## B.6 PROOF OF THEOREM 4.14 (EMPIRICAL BAYES OPTIMALITY VIA AUGMENTED ELBO)

*Proof.* The proof leverages variational inference principles. Start with the log marginal likelihood:

$$\log p(\mathcal{D}) = \log \int\int\int p(\mathcal{D},\boldsymbol{\theta},\phi,\gamma)\,d\boldsymbol{\theta}\,d\phi\,d\gamma. \tag{50}$$

Introduce the variational distribution $q(\boldsymbol{\theta},\phi) = q(\boldsymbol{\theta})q(\phi)$:

$$\log p(\mathcal{D}) = \mathbb{E}_{q(\boldsymbol{\theta},\phi)}\left[\log \frac{p(\mathcal{D},\boldsymbol{\theta},\phi)}{q(\boldsymbol{\theta},\phi)}\right] + \mathrm{KL}(q(\boldsymbol{\theta},\phi)\|p(\boldsymbol{\theta},\phi\,|\,\mathcal{D})). \tag{51}$$

Since the KL divergence is non-negative, we obtain the lower bound:

$$\log p(\mathcal{D}) \geq \mathbb{E}_{q(\boldsymbol{\theta},\phi)}\left[\log \frac{p(\mathcal{D},\boldsymbol{\theta},\phi)}{q(\boldsymbol{\theta},\phi)}\right]. \tag{52}$$

Now, factor in the augmentation variable $\boldsymbol{\gamma}$. The joint likelihood is:

$$p(\mathcal{D} \mid \boldsymbol{\theta}, \boldsymbol{\phi}) = \mathbb{E}_{p(\boldsymbol{\gamma} \mid \boldsymbol{\phi})} \left[ \prod_{i=1}^{N} p(\mathbf{y}_i \mid T_{\boldsymbol{\gamma}}(\mathbf{x}_i), \boldsymbol{\theta}) \right]. \tag{53}$$

Applying Jensen's inequality:

$$\log p(\mathcal{D} \mid \boldsymbol{\theta}, \boldsymbol{\phi}) \geq \mathbb{E}_{p(\boldsymbol{\gamma} \mid \boldsymbol{\phi})} \left[ \sum_{i=1}^{N} \log p(\mathbf{y}_i \mid T_{\boldsymbol{\gamma}}(\mathbf{x}_i), \boldsymbol{\theta}) \right]. \tag{54}$$

Substitute into the lower bound:

$$\log p(\mathcal{D}) \geq \mathbb{E}_{q(\boldsymbol{\theta}, \boldsymbol{\phi})} \left[ \mathbb{E}_{p(\boldsymbol{\gamma} \mid \boldsymbol{\phi})} \left[ \sum_{i=1}^{N} \log p(\mathbf{y}_i \mid T_{\boldsymbol{\gamma}}(\mathbf{x}_i), \boldsymbol{\theta}) \right] + \log \frac{p(\boldsymbol{\theta}) p(\boldsymbol{\phi})}{q(\boldsymbol{\theta}) q(\boldsymbol{\phi})} \right]. \tag{55}$$

Rewrite:

$$= \mathbb{E}_{q(\boldsymbol{\theta})} \mathbb{E}_{q(\boldsymbol{\phi})} \mathbb{E}_{p(\boldsymbol{\gamma} \mid \boldsymbol{\phi})} \left[ \sum_{i=1}^{N} \log p(\mathbf{y}_i \mid T_{\boldsymbol{\gamma}}(\mathbf{x}_i), \boldsymbol{\theta}) \right] - \mathrm{KL}(q(\boldsymbol{\theta}) \| p(\boldsymbol{\theta})) - \mathrm{KL}(q(\boldsymbol{\phi}) \| p(\boldsymbol{\phi})). \tag{56}$$

Thus, we arrive at the augmented ELBO. When $q(\boldsymbol{\theta}) = p(\boldsymbol{\theta} \mid \mathcal{D})$ and $q(\boldsymbol{\phi}) = p(\boldsymbol{\phi} \mid \mathcal{D})$, the bound becomes tight, confirming the result. $\qquad \square$

### B.7 PROOF OF THEOREM 4.17 (INFORMATION GAIN FROM AUGMENTATION)

*Proof.* Under a Gaussian approximation to the posterior, the entropy is proportional to the log determinant of the covariance matrix. Using the results from Theorem Theorem 4.12, we have:

$$\Delta H \propto \log \det(\Sigma_{\mathrm{noaug}}) - \log \det(\Sigma_{\mathrm{aug}}), \tag{57}$$

where $\Sigma_{\mathrm{noaug}} \approx H_{\mathrm{noaug}}^{-1}$, $\Sigma_{\mathrm{aug}} \approx H_{\mathrm{aug}}^{-1}$, and $H_{\mathrm{aug}}$ incorporates the effect of augmentation. Thus:

$$\Delta H \approx \frac{1}{2} \log \det(H_{\mathrm{aug}} H_{\mathrm{noaug}}^{-1}) = \frac{1}{2} \log \det(I + H_{\mathrm{noaug}}^{-1}(H_{\mathrm{aug}} - H_{\mathrm{noaug}})). \tag{58}$$

Approximating the effect of augmentation as an effective increase in Fisher information, we obtain the stated result. $\qquad \square$

## C A PRIMER ON PAC-BAYES THEORY

The Probably Approximately Correct (PAC)-Bayes framework, pioneered by (McAllester, 1999) and further developed by (Catoni, 2007) among others, provides a powerful tool for deriving generalization bounds for Bayesian-inspired learning algorithms. Unlike traditional PAC learning which often focuses on a single hypothesis, PAC-Bayes theory considers a distribution over hypotheses.

**Core Idea** The central idea is to bound the true risk (expected loss on unseen data) of a *posterior* distribution $Q$ over a hypothesis class $\mathcal{H}$. This bound is typically expressed in terms of the empirical risk (average loss on the training data) under $Q$, and a complexity term that measures how much $Q$ deviates from a data-independent *prior* distribution $P$ over $\mathcal{H}$. The guarantee holds with high probability (at least $1 - \delta$) over the random draw of the training dataset.

**Key Components**

- **Hypothesis Class** ($\mathcal{H}$): The set of all possible models (e.g., sets of parameters $\boldsymbol{\theta}$).
- **Prior Distribution** ($P$): A distribution over $\mathcal{H}$ chosen *before* observing any training data. It reflects initial beliefs about good hypotheses.
- **Posterior Distribution** ($Q$): A distribution over $\mathcal{H}$ that is typically learned from the training data $\mathcal{D}$. In PAC-Bayes, $Q$ can be any distribution, not necessarily a true Bayesian posterior.

- **Loss Function ($\ell(h, z)$):** Measures the error of a hypothesis $h \in \mathcal{H}$ on a data point $z = (\mathbf{x}, \mathbf{y})$.

- **True Risk ($R(Q)$):** The expected loss of a hypothesis drawn from $Q$ on the true (unknown) data distribution: $R(Q) = \mathbb{E}_{h \sim Q}[\mathbb{E}_{z \sim \mathcal{D}_{\text{true}}}[\ell(h, z)]]$.

- **Empirical Risk ($\hat{R}(Q)$):** The average loss of a hypothesis drawn from $Q$ on the $N$ training samples: $\hat{R}(Q) = \mathbb{E}_{h \sim Q}[\frac{1}{N} \sum_{i=1}^{N} \ell(h, z_i)]$.

- **Kullback-Leibler (KL) Divergence ($\mathbf{KL}(Q\|P)$):** Measures the "distance" or "information gain" from the prior $P$ to the posterior $Q$. It serves as a complexity penalty: if $Q$ is very different from $P$, the penalty is high.

**A Common Form of PAC-Bayes Bound**  A typical PAC-Bayes generalization bound (e.g., McAllester's 1999 bound or variations) states that for any $\delta \in (0, 1)$, with probability at least $1 - \delta$ over the draw of an i.i.d. training set $\mathcal{D}$ of size $N$, for all posterior distributions $Q$:

$$R(Q) \leq \hat{R}(Q) + \sqrt{\frac{\text{KL}(Q\|P) + \ln(\frac{1}{\delta}) + C}{2N}} \tag{59}$$

where $C$ is a constant that can depend on the range of the loss or other factors (e.g., $\ln(2\sqrt{N})$ as used in our paper, which is a common variant for empirical Bernstein bounds).

**Interpretation and Significance**

- The bound guarantees that the true risk is unlikely to be much larger than the empirical risk, plus a term that penalizes the complexity of $Q$ relative to $P$.

- It highlights a trade-off: to achieve good generalization, a learning algorithm should find a posterior $Q$ that both fits the training data well (low $\hat{R}(Q)$) and does not deviate too much from the prior (low $\text{KL}(Q\|P)$).

- The bounds are often tighter than uniform convergence bounds for complex hypothesis classes like neural networks, especially when a good prior is available.

- They provide a theoretical justification for regularization techniques and can guide the design of learning algorithms.

**Relevance to This Paper**  In our work (§ 4.2), we adapt this framework to derive generalization bounds for our augmented likelihood approach. Here, the "hypothesis" space effectively includes both the model parameters $\boldsymbol{\theta}$ and the augmentation (hyper)parameters $\boldsymbol{\phi}$. The priors $p(\boldsymbol{\theta})$ and $p(\boldsymbol{\phi})$ and the variational posteriors $q(\boldsymbol{\theta})$ and $q(\boldsymbol{\phi})$ play the roles of $P$ and $Q$. Our Theorem Theorem 4.4 provides such a bound, and Theorem Theorem 4.5 uses PAC-Bayes reasoning to show the theoretical advantage of our marginalized approach over naïve data replication. The KL terms in our augmented ELBO (Eq. 7) naturally appear as complexity measures in these PAC-Bayes bounds.

# D  ALGORITHM AND IMPLEMENTATION

We now present a practical algorithm for implementing our Bayesian-optimized data augmentation approach. The algorithm employs stochastic gradient-based optimization of both model parameters and augmentation distribution parameters.

## D.1  PARAMETERIZATION OF AUGMENTATION DISTRIBUTION

For continuous transformation parameters, we typically use a Gaussian distribution for $p(\boldsymbol{\gamma} \,|\, \boldsymbol{\phi})$:

$$p(\boldsymbol{\gamma}|\boldsymbol{\phi}) = \mathcal{N}(\boldsymbol{\gamma}|\mu_{\boldsymbol{\phi}}, \Sigma_{\boldsymbol{\phi}}), \tag{60}$$

where $\boldsymbol{\phi} = (\mu_{\boldsymbol{\phi}}, \Sigma_{\boldsymbol{\phi}})$. For $q(\boldsymbol{\phi})$, we might use a Gaussian:

$$q(\boldsymbol{\phi}) = \mathcal{N}(\boldsymbol{\phi} \,|\, \mu_q, \Sigma_q), \tag{61}$$

---

**Algorithm 1** Augmented Variational Inference with Learned Augmentation

---

1: **Input:** Dataset $\mathcal{D} = \{(\mathbf{x}_i, \mathbf{y}_i)\}_{i=1}^N$, transformation family $T_{\boldsymbol{\gamma}}(\cdot)$
2: **Initialize:** Variational distributions $q(\boldsymbol{\theta})$ and $q(\boldsymbol{\phi})$
3: **while** not converged **do**
4:     Sample a minibatch $\{(\mathbf{x}_i, \mathbf{y}_i)\}_{i=1}^B$ from $\mathcal{D}$
5:     Sample model parameters $\boldsymbol{\theta} \sim q(\boldsymbol{\theta})$ (or use reparameterization)
6:     Sample augmentation parameters $\boldsymbol{\phi} \sim q(\boldsymbol{\phi})$ (or use reparameterization)
7:     **for** each $(\mathbf{x}_i, \mathbf{y}_i)$ in the minibatch **do**
8:         Sample augmentation parameters $\boldsymbol{\gamma}_i \sim p(\boldsymbol{\gamma}|\boldsymbol{\phi})$
9:         Apply transformation $\mathbf{x}_i' = T_{\boldsymbol{\gamma}_i}(\mathbf{x}_i)$
10:        Compute log-likelihood $\log p(\mathbf{y}_i|\mathbf{x}_i', \boldsymbol{\theta})$
11:    **end for**
12:    Estimate ELBO:

$$\widehat{\text{ELBO}}_{\text{aug}} = \frac{N}{B} \sum_{i=1}^B \log p(\mathbf{y}_i|\mathbf{x}_i', \boldsymbol{\theta}) - \text{KL}(q(\boldsymbol{\theta})||p(\boldsymbol{\theta})) - \text{KL}(q(\boldsymbol{\phi})||p(\boldsymbol{\phi}))$$

13:    Update variational parameters in $q(\boldsymbol{\theta})$ using gradient of $\widehat{\text{ELBO}}_{\text{aug}}$
14:    Update variational parameters in $q(\boldsymbol{\phi})$ using gradient of $\widehat{\text{ELBO}}_{\text{aug}}$
15: **end while**
16: **Output:** Optimized variational distributions $q(\boldsymbol{\theta})$ and $q(\boldsymbol{\phi})$

---

**Algorithm 2** Partial Variational Inference with Learned Augmentation

---

1: **Input:** Dataset $\mathcal{D} = \{(\mathbf{x}_i, \mathbf{y}_i)\}_{i=1}^N$, transformation family $T_{\boldsymbol{\gamma}}(\cdot)$
2: **Initialize:** Model parameters $\boldsymbol{\theta}$ and distribution $q(\boldsymbol{\phi})$
3: **while** not converged **do**
4:     Sample a minibatch $\{(\mathbf{x}_i, \mathbf{y}_i)\}_{i=1}^B$ from $\mathcal{D}$
5:     Sample augmentation parameters $\boldsymbol{\phi} \sim q(\boldsymbol{\phi})$
6:     **for** each $(\mathbf{x}_i, \mathbf{y}_i)$ in the minibatch **do**
7:         Sample $\boldsymbol{\gamma}_i \sim p(\boldsymbol{\gamma}|\boldsymbol{\phi})$
8:         Apply transformation $\mathbf{x}_i' = T_{\boldsymbol{\gamma}_i}(\mathbf{x}_i)$
9:         Compute log-likelihood $\log p(\mathbf{y}_i|\mathbf{x}_i', \boldsymbol{\theta})$
10:    **end for**
11:    Estimate ELBO:

$$\widehat{\text{ELBO}}_{\text{aug}} = \frac{N}{B} \sum_{i=1}^B \log p(\mathbf{y}_i|\mathbf{x}_i', \boldsymbol{\theta}) - \text{KL}(q(\boldsymbol{\phi})||p(\boldsymbol{\phi}))$$

12:    Update $\boldsymbol{\theta}$ using gradient of $\widehat{\text{ELBO}}_{\text{aug}}$
13:    Update $q(\boldsymbol{\phi})$ using gradient of $\widehat{\text{ELBO}}_{\text{aug}}$
14: **end while**
15: **Output:** Optimized parameters $\boldsymbol{\theta}$ and distribution $q(\boldsymbol{\phi})$

---

learning $\mu_q$ and $\Sigma_q$. This allows for reparameterization during sampling:

$$\boldsymbol{\phi} = \mu_q + \Sigma_q^{1/2}\epsilon, \quad \epsilon \sim \mathcal{N}(0, I), \tag{62}$$

followed by:

$$\boldsymbol{\gamma} = \mu_{\boldsymbol{\phi}} + \Sigma_{\boldsymbol{\phi}}^{1/2}\epsilon', \quad \epsilon' \sim \mathcal{N}(0, I). \tag{63}$$

For discrete transformations, we can use a categorical distribution:

$$p(\boldsymbol{\gamma}|\boldsymbol{\phi}) = \text{Cat}(\boldsymbol{\gamma}|\pi_{\boldsymbol{\phi}}), \tag{64}$$

where $\pi_{\boldsymbol{\phi}}$ represents the probabilities, and use the Gumbel-Softmax trick (Jang et al., 2017) for differentiable sampling.

## D.2 Practical Considerations

**Adaptive Variance Scheduling.**   Based on Corollary 4.2, we can implement an adaptive schedule for the augmentation variance within $q(\phi)$, adjusting the variance of $\phi$ over training to balance exploration and bound tightness.

**Marginalization Advantage Monitoring.**   Following Corollary 4.6, we can monitor the marginalization advantage term $D_\phi(x_i, y_i)$ during training to assess the benefit of OPTIMA over naïve augmentation.

**Curvature-Aware Augmentation.**   Inspired by Corollary 4.10, we can adapt the augmentation distribution based on the model's sensitivity to different transformations, allocating more variance to directions where the model is approximately invariant.

**Computational Efficiency.**   For large models, we use Monte Carlo estimates with a small number of samples (e.g., one per data point per iteration) to approximate the expectations in the ELBO. The reparameterization trick ensures low-variance gradient estimates.

# E   Additional Experimental Details for § 5.2

We use a ResNet-50 architecture with a Bayesian linear layer at the end (for non-Bayesian case, we just use ResNet-50 without any replacements). We apply standard preprocessing for IMAGENET. We use the Adam optimizer with a learning rate of $1 \times 10^{-5}$ for model parameters. In OPTIMA, we are learning a parameter in Beta distribution for Mixup, in uniform distribution for Cutmix, and Dirichlet, depth and Beta distribution parameters jointly for Augmix augmentations. For these augmentations, we use lognormal distribution as a prior because of the simplicity. The augmentation parameters have a separate learning rate ($1 \times 10^{-3}$) to facilitate faster exploration. We regularize the augmentation parameters with a KL weight of $\beta_{\text{kl\_aug}} = 1$, balancing data-fit and prior alignment. For all methods, we include a small KL weight $\beta_{\text{kl\_net}} = 10^{-4}$ on the model parameters to maintain a Bayesian prior but it works with any weight on the model parameters This acts as a Bayesian regularizer on the final layer weights, preventing overfitting within that layer and ensuring consistency with the variational Bayesian framework (Blundell et al., 2015). Training proceeds for 30 epochs with a batch size of 256.

**Evaluation Metrics.**   We compute the Expected Calibration Error (ECE) by dividing predictions into 10 bins based on confidence and measuring the difference between average confidence and accuracy in each bin:

$$\text{ECE} = \sum_{i=1}^{10} \frac{|B_i|}{n} |\text{acc}(B_i) - \text{conf}(B_i)|, \tag{65}$$

where $B_i$ is the set of examples in bin $i$, $n$ is the total number of examples, $\text{acc}(B_i)$ is the accuracy in bin $i$, and $\text{conf}(B_i)$ is the average confidence in bin $i$.

For out-of-distribution detection, we use the AUROC metric, which measures the area under the ROC curve when using predictive entropy as the detection score:

$$H[p(y|x)] = -\sum_{c=1}^{C} p(\mathbf{y} = c \,|\, \mathbf{x}) \log p(\mathbf{y} = c \,|\, \mathbf{x}). \tag{66}$$

Higher entropy indicates higher uncertainty, which should correlate with out-of-distribution examples.

# F   Additional Results on Different Types of Data Augmentation

## F.1 Learning Geometric Augmentation for cifar10 Classification

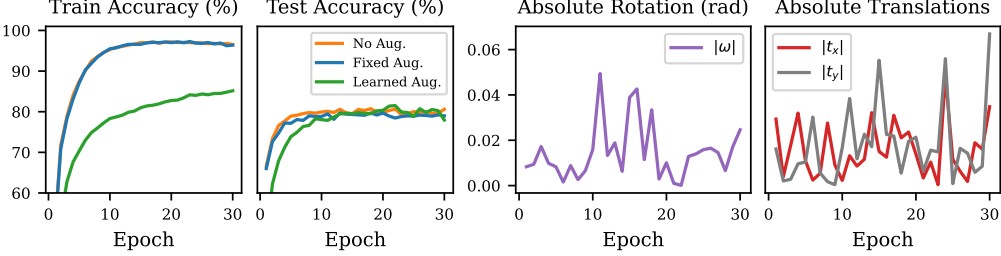

Figure 3: *(Left Two)* Convergences of training and test accuracy on CIFAR10. OPTIMA generelizes better than the other approaches. *(Right Two)* Evolutions of the data augmentation parameters.

**Setups.** We use a ResNet-18 architecture with a Bayesian linear layer at the end. We apply standard preprocessing: normalization with mean (0.4914, 0.4822, 0.4465) and standard deviation (0.2023, 0.1994, 0.2010). We use the Adam optimizer with a learning rate of $1 \times 10^{-4}$ for model parameters. In OPTIMA, we are learning $\gamma = \{\omega, t_x, t_y\}$ jointly, where $\omega$ is rotation (radians), and $t_x$ and $t_y$ are horizontal and vertical shifts. The augmentation parameters have a separate learning rate ($1 \times 10^{-2}$) to facilitate faster exploration. We regularize the augmentation parameters with a KL weight of $\beta_{\text{kl\_aug}} = 1$,

Table 5: CIFAR10 classification. Fixed Aug uses fixed rotation $\omega = 0.1$ and translations of $0.1$.

| Method | Acc (%) | ECE ($\downarrow$) |
|---|---|---|
| No Aug | 80.90 | 0.092 |
| Fixed Aug | 80.73 | 0.088 |
| OPTIMA | **81.35** | **0.017** |

balancing data-fit and prior alignment. For all methods, we include a small KL weight $\beta_{\text{kl\_net}} = 0.1$ on the model parameters to maintain a Bayesian prior but it works with any weight on the model parameters. This acts as a Bayesian regularizer on the final layer weights, preventing overfitting within that layer and ensuring consistency with the variational Bayesian framework (Blundell et al., 2015). Training proceeds for 30 epochs with a batch size of 128.

**Results.** We assess calibration using 100 Monte Carlo (MC) samples. Fig. 1 presents reliability diagrams for **No Aug**, **Fixed Aug**, and OPTIMA, revealing that the learned augmentation strategy yields the lowest calibration error (ECE), with the reliability curve closely aligning with perfect calibration. Table 5 summarizes the final ECE values, confirming that OPTIMA leads to more accurate confidence estimates than fixed or no augmentation. Moreover, Fig. 3 (second panel) shows test accuracy over time: the learned augmentation generalizes better, while **No Aug** and **Fixed Aug** exhibit overfitting and poorer generalization.

## F.2 EXPLORING DIFFERENT INTENSITIES OF GAUSSIAN TRANSLATIONS

We use the same implementation details as in Appendix F.1, except that we choose Gaussian Translation as an augmentation parameter and validate on different values of $K$ in naïve augmentation, and different prior variance $\sigma$ in OPTIMA. As an OOD data, we choose the SVHN dataset, since this mismatch makes it a widely adopted benchmark for OOD testing of classifiers trained on CIFAR10.

Table 6: Effect of marginalization vs. naïve augmentation with different numbers of augmentations per example on CIFAR10 using Pretrained Bayesian ResNet-18 (last layer) and Gaussian Translation after 30 epochs. Test accuracy and ECE are on CIFAR10, and OOD AUROC is on the SVHN dataset.

| Method | Test Acc (%) | ECE $\downarrow$ | OOD AUROC |
|---|---|---|---|
| No Aug | 94.09 | 0.0381 | 0.9069 |
| naïve Aug (K=2) | 95.03 | 0.0298 | 0.9425 |
| naïve Aug (K=5) | 95.21 | 0.0327 | 0.9383 |
| naïve Aug (K=10) | 93.75 | 0.0424 | 0.9560 |
| OPTIMA ($\sigma = 0.1$) | 93.30 | 0.0192 | 0.9446 |
| OPTIMA ($\sigma = 0.5$) | 93.87 | **0.0165** | **0.9576** |
| OPTIMA ($\sigma = 1$) | 90.25 | **0.0175** | **0.9647** |

**Results.** Table 6 shows that OPTIMA allows us to get much better calibration than naïve and no augmentation cases. Because of the overcounting problem in naïve case, it obviously consumes

around K times more time than our approach demostrating that we can get good generalization and better robustness for OOD in short time with our approach.

## G   Additional Experiment on ImageNet using ResNet-50

Here we use OPTIMA for Imagenet in order to learn Mixup, Cutmix and Augmix augmentations.

### G.1   Implementation Details for OPTIMA with Mixup

We evaluate our OPTIMA framework with the Learnable Mixup augmentation on the ImageNet (Deng et al., 2009) dataset for image classification. Performance is assessed on the standard ImageNet validation set, and robustness is measured on the ImageNet-C (Hendrycks & Dietterich, 2019) benchmark.

**Model Architecture and Preprocessing.** We employ a ResNet-50 architecture (He et al., 2016), initialized with pretrained weights from `torchvision.models.ResNet50_Weights.IMAGENET1K_V2`. The final fully connected layer is replaced with a new linear layer mapping to the 1000 ImageNet classes. For input preprocessing during training, images are transformed by a `RandomResizedCrop` to $224 \times 224$ pixels followed by a `RandomHorizontalFlip`. Validation and test images are resized to 256 pixels on their shorter edge and then center-cropped to $224 \times 224$. All images are subsequently converted to tensors and normalized using the standard ImageNet mean $\boldsymbol{\mu}_{\text{ImageNet}} = (0.485, 0.456, 0.406)$ and standard deviation $\boldsymbol{\sigma}_{\text{ImageNet}} = (0.229, 0.224, 0.225)$.

**Learnable Mixup Augmenter.**   In the OPTIMA Mixup variant, the Mixup hyperparameter $\alpha$ (controlling the Beta distribution $\text{Beta}(\alpha, \alpha)$ from which the mixing coefficient $\lambda$ is sampled) is made learnable. We parameterize a Normal distribution over $\text{logit}(\alpha)$ with learnable mean $\mu_{\ell\alpha}$ and learnable log standard deviation $\log \sigma_{\ell\alpha}$, where $\ell\alpha = \text{logit}(\alpha)$. The initial value for $\mu_{\ell\alpha}$ is set to $\text{logit}(0.2)$, corresponding to an initial $\alpha_{\text{init}} = 0.2$. The initial $\log \sigma_{\ell\alpha}$ is set to $\log(0.1)$, promoting a small initial variance for the learned distribution over $\text{logit}(\alpha)$. A prior distribution $p(\text{logit}(\alpha))$ is defined as $\mathcal{N}(\text{logit}(\alpha_{\text{init}}), \sigma_p^2)$, where the prior standard deviation $\sigma_p = 2.0$. The KL divergence between the learned variational posterior $q(\text{logit}(\alpha)|\mu_{\ell\alpha}, \sigma_{\ell\alpha}^2)$ and this prior is added to the training objective, weighted by the hyperparameter `beta_augmenter_reg`. Sampled $\lambda$ values are clamped to the range $[10^{-6}, 1 - 10^{-6}]$ for numerical stability.

**Training Configuration.**   Models were trained for 10 epochs[1] using the AdamW optimizer. The base learning rate for the ResNet-50 parameters was set to $1 \times 10^{-4}$. The learnable parameters of the Mixup augmenter ($\mu_{\ell\alpha}, \log \sigma_{\ell\alpha}$) utilized a learning rate of $1 \times 10^{-3}$ (10 times the base learning rate). A cosine annealing learning rate scheduler with warm restarts (`CosineAnnealingWarmRestarts`) was employed, with parameters '$T_0 = 10$' epochs, '$T_{mult} = 2$', and '$eta_{min}$' set to $1/100$ of the initial learning rate. The weight decay for network parameters (`beta_network_reg`) was 0.01. The coefficient for the KL divergence term of the augmentation parameters (`beta_augmenter_reg`) was 1.0. Training was performed with a global batch size of 256 distributed across 4 GPUs using Distributed Data Parallel (DDP). We used a precision of '"16"' (interpreted as 16-bit native mixed precision) and set '$torch.set\_float32\_matmul\_precision('medium')$'. Gradient clipping was applied with a maximum norm of 1.0. The number of data loader workers was set to 8 per GPU process.

**Baselines.**   We compare OPTIMA Mixup against:

- **Fixed Mixup**: Standard Mixup augmentation with a fixed $\alpha = 0.2$. The training setup (optimizer, scheduler, epochs, batch size) was identical to that of OPTIMA Mixup, excluding elements specific to learnable augmentation parameters.

---

[1]While longer training (e.g., 90-100 epochs) is standard for ImageNet, these experiments were conducted for 10 epochs to demonstrate the behavior of the learnable augmentation parameters and compare against fixed augmentation under identical short-run conditions.

- **Pretrained ResNet-50 (No Augmentation Eval)**: The ResNet-50 model with weights from `torchvision.models.ResNet50_Weights.IMAGENET1K_V2`, evaluated directly on the validation and IMAGENET-C sets without any fine-tuning under our experimental setup. This serves as a standard reference.

### G.2 IMPLEMENTATION DETAILS FOR OPTIMA WITH CUTMIX

For evaluating OPTIMA with CutMix, we follow a similar experimental setup on the IMAGENET (Deng et al., 2009) dataset, with robustness assessed on IMAGENET-C (Hendrycks & Dietterich, 2019).

**Model Architecture and Preprocessing.** We use the ResNet-50 architecture (He et al., 2016) pretrained with `torchvision.models.ResNet50_Weights.IMAGENET1K_V2`, replacing the final classifier layer for the 1000 IMAGENET classes. Input preprocessing during training includes `RandomResizedCrop` to $224\times224$ and `RandomHorizontalFlip`. Validation and test images are resized (256 shorter edge) and center-cropped to $224 \times 224$. Standard IMAGENET normalization is applied.

**Learnable CutMix Augmenter.** In CutMix (Yun et al., 2019), a patch from one image is pasted onto another, and labels are mixed proportionally to the area of the patches. The mixing ratio $\lambda$ (determining the area of the first image to keep, and thus (1-$\lambda$) is the area of the patch from the second image) is typically sampled from a Beta$(\alpha, \alpha)$ distribution. For our OPTIMA CutMix, this $\alpha$ parameter of the Beta distribution is made learnable. We parameterize a Normal distribution over $\log(\alpha)$ with learnable mean $\mu_{\log \alpha}$ and learnable log standard deviation $\log \sigma_{\log \alpha}$. The initial value for $\mu_{\log \alpha}$ is set to $\log(1.0)$, corresponding to an initial $\alpha_{\text{init}} = 1.0$ (a common default for CutMix). The initial $\log \sigma_{\log \alpha}$ is set to $\log(0.1)$. A prior distribution $p(\log(\alpha))$ is defined as $\mathcal{N}(\log(\alpha_{\text{init}}), \sigma_p^2)$, with prior standard deviation $\sigma_p = 2.0$. The KL divergence between the learned variational posterior for $\log(\alpha)$ and this prior is added to the training loss, weighted by `beta_augmenter_reg`. The sampled $\alpha$ values are clamped to $[10^{-4}, 100.0]$ before being used in the Beta distribution. The resulting mixing coefficient $\lambda_{\text{final}}$ (coefficient for the first image's label) is determined by the actual area of the pasted patch after clipping to image boundaries.

**Training Configuration.** Models were trained for $N_{\text{epochs}}$ epochs (e.g., 15) using the AdamW optimizer. The base learning rate for network parameters was $1\times10^{-4}$, while the learnable CutMix parameters ($\mu_{\log \alpha}, \log \sigma_{\log \alpha}$) used a learning rate of $1 \times 10^{-3}$. A `CosineAnnealingWarmRestarts` learning rate scheduler was used ('T_0=10' or '15', 'T_mult=2', 'eta_min'=$^1/_{100}$ of initial LR). Network weight decay (`beta_network_reg`) was 0.01, and the KL coefficient (`beta_augmenter_reg`) was 1.0. Training used a global batch size of 256 on 4 GPUs with DDP, '"16"' precision, '$torch.set\_float32\_matmul\_precision('medium')$', and gradient clipping at 1.0. Data loader workers were set to 8 per GPU.

### G.3 IMPLEMENTATION DETAILS FOR OPTIMA WITH AUGMIX (LEARNABLE SEVERITY + JSD)

We evaluate OPTIMA by learning a component of the AugMix (Hendrycks et al., 2020) augmentation strategy, specifically its overall `aug_severity`, while also employing the Jensen-Shannon Divergence (JSD) consistency loss. Experiments are conducted on IMAGENET (Deng et al., 2009) and IMAGENET-C (Hendrycks & Dietterich, 2019).

**Model Architecture and Preprocessing.** The model is a ResNet-50 (He et al., 2016) initialized with `torchvision.models.ResNet50_Weights.IMAGENET1K_V2`, with the final classifier layer adapted for 1000 classes. During training, input PIL images undergo `RandomResizedCrop` to $224 \times 224$ and `RandomHorizontalFlip`. These PIL images are then passed to our `LearnableAugMixSeverityJSDAugmenter` module. Validation and test images use standard resizing, center cropping, and `ToTensor` conversion, followed by IMAGENET normalization.

**Learnable AugMix Severity + JSD Augmenter.** The `LearnableAugMixSeverityJSDAugmenter` is implemented as an `nn.Module`.

- **AugMix Core**: For each input PIL image, three views are generated: the original, and two independently augmented versions using AugMix. Each AugMix version is a convex combination ($m \sim \text{Beta}(1, 1)$) of the original image and a mixture of $K = $ augmix_mixture_width (default 3) augmentation chains. Each chain consists of $D$ (default random 1-3, controlled by augmix_mixture_depth) basic operations (e.g., rotate, shear, color jitter) sampled randomly. Mixing weights for chains $w_k$ are from Dirichlet($\mathbf{1}$). All PIL operations and the final conversion to tensors (for each of the three views) happen within this augmenter module, ensuring output tensors are on the correct device. We utilize a predefined list of tensor-based augmentation operations where possible to improve performance over PIL-only operations.
- **Learnable Severity**: The overall intensity of the basic augmentations, aug_severity (typically a value between 0-10), is made learnable. We parameterize a Normal distribution over $\log(\text{aug\_severity})$ with learnable mean $\mu_{\log S}$ and learnable log standard deviation $\log \sigma_{\log S}$. The initial $\mu_{\log S}$ corresponds to an initial_aug_severity of 3.0, and initial $\log \sigma_{\log S} = \log(0.1)$. The prior for $\log(\text{aug\_severity})$ is $\mathcal{N}(\log(\text{initial\_aug\_severity}), \sigma_{pS}^2)$ with $\sigma_{pS} = $ prior_severity_std_learnable_aug (default 1.0). A KL divergence term, weighted by beta_augmenter_reg, regularizes these learned severity parameters. The sampled severity is clamped to $[0.1, 10.0]$.
- **JSD Loss**: The three output image tensors (original, AugMix view 1, AugMix view 2) are passed through the network. A JSD consistency loss is calculated between their softmax predictions, weighted by beta_jsd (default 12.0), and added to the primary cross-entropy loss (calculated on the original view).

The data loader for training uses a custom collate function to provide a list of PIL images to the augmenter.

**Training Configuration.** Training was conducted for $N_{\text{epochs}}$ epochs (in our case 6 epochs because of the computational complexity related to tensor and PIL tranformations) with the AdamW optimizer. The base learning rate for network parameters was $1 \times 10^{-4}$. The learnable severity parameters ($\mu_{\log S}, \log \sigma_{\log S}$) used a learning rate of $1 \times 10^{-3}$. A CosineAnnealingWarmRestarts scheduler was used ('T_0=10' or '15', 'T_mult=2', 'eta_min'=$1/100$ of initial LR). Network weight decay (beta_network_reg) was 0.01. The KL coefficient for severity parameters (beta_augmenter_reg) was 1.0. The JSD loss coefficient (beta_jsd) was 12.0. Training used a global batch size of 128 (reduced due to processing three views) on 4 GPUs with DDP, '16' precision, '$torch.set\_float32\_matmul\_precision('medium')$', and gradient clipping at 1.0. Data loader workers were 8 per GPU.

## G.4 Evaluation and Software/Hardware for all these methods.

Models are evaluated on the standard IMAGENET validation set for top-1 accuracy and cross-entropy loss. Robustness is assessed on the IMAGENET-C benchmark, reporting the normalized mean Corruption Error (mCE normalized by AlexNet baseline) across all corruptions and severities. For IMAGENET-C, images are processed using the same validation transforms as for the clean IMAGENET validation set. All final reported evaluations are performed on a single GPU using 32-bit floating-point precision.

Experiments were conducted using PyTorch version 2.0.1 and PyTorch Lightning version 2.1.0. Training and evaluation four utilized GPUs.

## G.5 Experimental Results

Table 7: The result of IMAGENET and IMAGENET-C using ResNet-50 for each augmentations. We evaluate the average test error for each corruption type

| Method | Test Acc (%) | mCE (normalized) (%) | gaussian | shot | impulse | defocus | glass | motion | zoom | snow | frost | fog | brightness | contrast | elastic | pixelate | jpeg |
|---|---|---|---|---|---|---|---|---|---|---|---|---|---|---|---|---|---|
| No Aug | 60.8 | 76.7 | 71 | 73 | 76 | 61 | 73 | 61 | 64 | 67 | 62 | 54 | 32 | 61 | 55 | 55 | 47 |
| Mixup (Zhang et al., 2018) | 77.9 | - | - | - | - | - | - | - | - | - | - | - | - | - | - | - | - |
| OPTIMA Mixup (10 epochs) | **79.31** | 68.41 | 55 | 57 | 59 | 62 | 73 | 59 | 59 | 62 | 51 | 44 | 31 | 48 | 55 | 50 | 43 |
| Cutmix (Yun et al., 2019) | 78.6 | - | - | - | - | - | - | - | - | - | - | - | - | - | - | - | - |
| OPTIMA Cutmix (15 epochs) | **79.62** | 70.6 | 68 | 68.7 | 68.8 | 74.6 | 88.6 | 75.7 | 73.4 | 73.2 | 70.8 | 59 | 54.1 | 61.3 | 83.6 | 70.5 | 69.2 |
| Augmix (Hendrycks et al., 2020) | 77.53 | 65.3 | - | - | - | - | - | - | - | - | - | - | - | - | - | - | - |
| OPTIMA Augmix (6 epochs) | **78.19** | 68.21 | 57.37 | 58.26 | 60.79 | 60.98 | 72.04 | 56.77 | 54.54 | 59.76 | 54.58 | 44.72 | 30.13 | 45.37 | 54.75 | 51.66 | 44.71 |

**Results.**    In Table 7, we can see that OPTIMA allows us to get better test accuracy on clean data with non-Bayesian ResNet-50. OPTIMA Mixup, Cutmix and Augmix are beating the baseline results within (5-15 epochs). The mCE of OPTIMA Augmix is lower than the benchmark. This can be explained by the very few training epochs (6 epochs) which we could run due to the computational complexity of this experiment; with our computational resources it takes around 15 hours for one full epoch.

## H  Additional Experimental Details for § 5.4. Token-dropout Implementation Details

**Parameterization.**    The augmentation module applies token dropout with probability

$$p_{\mathrm{drop}} = p_{\max}\, \sigma(s),$$

where $s$ is a trainable scalar (initialized at $s_0 = -2$ in our implementation) and $\sigma(\cdot)$ is the logistic function. The constant $p_{\max}$ sets an upper bound on the amount of dropout; we use $p_{\max} = 0.5$.

**Prior.**    We place a Gaussian prior directly on $p_{\mathrm{drop}}$,

$$p(p_{\mathrm{drop}}) \propto \exp\left(-\tfrac{1}{2}\tfrac{(p_{\mathrm{drop}}-\mu)^2}{\sigma^2}\right),$$

implemented as a quadratic penalty in the ELBO objective. We use $\mu \in \{0.1, 0.3\}$ to represent weak and strong prior preferences for token dropout, and $\sigma = 0.1$.

**OPTIMA initialization.**    All methods begin from the same initial dropout rate $p_{\mathrm{drop}} = p_{\max}\, \sigma(s_0) \approx 0.04$, ensuring a mild initial augmentation.

**Baselines.**    We compare the following: (i) *No Aug* ($p_{\mathrm{drop}} = 0$); (ii) *Fixed Aug*, using the same initialization as OPTIMA; (iii) *Fixed Aug (Matched)*, using OPTIMA's learned dropout; (iv) *BO-Fixed*, selecting $p_{\mathrm{drop}}$ via validation NLL over a grid in $[0, 0.3]$.

**Optimizers.**    DistilBERT is trained with learning rate $2 \times 10^{-5}$, and $s$ is trained with learning rate $5 \times 10^{-2}$. Further hyperparameters are unchanged from standard HuggingFace defaults.

## I  Broader Impact

OPTIMA has significant potential for improving the reliability of Bayesian deep learning in high-stakes applications, such as medical imaging, autonomous driving, and scientific discovery. The ability to learn optimal augmentation strategies from data also reduces the need for manual tuning, making Bayesian methods more accessible to practitioners across domains.

## J  Reproducibility Statement

All experiments are fully described in the submission, including dataset details, hyperparameters, and training procedures. The accompanying code is provided to ensure that our results can be independently reproduced.

## K  The Use of Large Language Models (LLMs)

We used large language models (LLMs) solely for non-substantive assistance, including grammar refinement and summarizing relevant literature. All research ideas, analyses, and conclusions are the authors' own.

