# OpenReview forum: "Optimizing Data Augmentation through Bayesian Model Selection"
_ICLR.cc/2026/Conference — ICLR 2026 Poster_

### Official Review · Reviewer_8dA2 · 2025-10-28

**Soundness:** 4
**Presentation:** 3
**Contribution:** 3
**Rating:** 8
**Confidence:** 4

**Summary:**

The paper introduces a Bayesian model selection framework for online optimization of the data augmentation (DA) hyperparameters, provides a tractable ELBO for variational approximation of the marginalized likelihood, performs extensive analysis of the theoretical characteristics of the augmented model, uncertainty quantification, calibration, decision boundary smoothness and generalization. The theoretical results are backed by several experiments with synthetic and image datasets optimizing the DA parameters of simple and also state of the art baselines in DA and improving the prediction and calibration with almost no additional computational cost.

**Strengths:**

The paper is well written and easy to follow. The authors have laid out the background and  the motivation and provide good structured explanation of the proposed method and its theoretical characteristics. Several experiments are provided to compare the Bayesian DA's performance to other baselines and highlight the theoretical results.

The authors use DA hyperparameters as latent mode parameters and use a Bayesian framework with ELBO optimization to efficiently find the best DA parameters according to the data invariance and model sensitivity. In the theoretical analysis, Bayesian DA method is compared to naïve DA in terms of the PAC-Bayes generalization bounds, regularization and smoothening effect of Bayesian DA and improvement in calibration of the predictions.

The analysis also sheds light on the connections of DA with information bottleneck which I find particularly novel.

**Weaknesses:**

It might be easier to follow the paper if a brief explanation of the theoretical results of the Theorems are provided before the theorems, as there are multiple theorems and moving from one technical result to the next requires some context in between.

Minor comment:
Typo: line 107 wrt -> with respect to
Typo: line 119 nav̈e -> naive
Typo: line 229 extra s

**Questions:**

Would the proposed method also work for when transformations are not continuous as in natural language e.g. based on word substitutions and syntactic transformations?

Can the authors provide the intuition of THM4.5? In the the naïve DA the variable $gamma$ is integrated out in $R(\theta)$ with a uniform empirical distribution as opposed to the latent distribution $p(\gamma) = \int p(\gamma|\phi)p(\phi|\theta)d\phi$ or its variational approximation $p(\gamma) = \int p(\gamma|\phi)q(\phi)d\phi$?

---

> ### Author Response · Authors · 2025-11-21
>
> Thank you for your highly positive review. We address your questions below.
>
> **1. Intuition before the theorems**
>
> We appreciate the suggestion. Due to the strict 10-page limit, we were unable to add further explanatory paragraphs directly in the main text.
>
> However, the intuitive roles of each theorem are already reflected in the surrounding discussion:
> - Section 4.1 motivates why variational marginalization approximates the true posterior and how this affects the model’s sensitivity to transformations.
>
>
> - Section 4.2 explains how PAC-Bayes theory connects Bayesian augmentation to generalization.
>
>
> - Section 4.3 links the smoothness of transformation families to contraction in output variance.
>
>
> - Sections 4.4–4.6 describe how naïve augmentation differs from marginalization in terms of variance, empirical Bayes interpretation, and entropy reduction.
>
>
> We hope this provides the conceptual intuition you were looking for, even without additional space for expanded pre-theorem summaries.
>
> **2. Discrete / non-continuous transformations**
>
> Thank you for raising this point. OPTIMA naturally handles discrete transformations because the augmentation parameters are treated as latent random variables, and their gradients can be estimated with score-function estimators when the transformations are non-differentiable.
>
> We added a new experiment on the NLP example in Section 5.4. Our experiment on SST-5 with token dropout illustrates this behavior: OPTIMA successfully learns a posterior over discrete deletion probabilities using this estimator. This demonstrates that the framework is not restricted to continuous transformations.
>
> **3. Intuition for Theorem 4.5**
>
> The key idea behind Theorem 4.5 is that naïve data augmentation approximates the effect of marginalizing over transformations, but does so through a finite-sample Monte Carlo estimator.
> - In naïve data augmentation, we draw K independent augmented samples and average the corresponding losses.
>   - This approximates the marginalized risk, but it introduces sampling noise, since we only average over a finite number of draws.
>
>
> - In contrast, Bayesian marginalization integrates out the augmentation variable analytically, using the latent distribution induced by the model. This eliminates Monte Carlo noise entirely.
>
>
> Thus, naïve DA is a noisy estimator of the marginalized objective, while the Bayesian formulation is the exact expectation.
>
> Theorem 4.5 formalizes this: the risk under naïve DA converges to the marginalized risk only as $K→\infty$, and the discrepancy is driven by the finite-sample randomness in naïve augmentation.
>
> This explains why marginalization leads to smoother decision boundaries and lower predictive uncertainty: it removes the instability caused by sampling a small number of augmented versions of each input.
>
> We highlighted changes in the revised version of the paper.
>
> We appreciate your positive assessment and hope the strengthened experimental part and clarified mathematical statements address your remaining concerns.

---

### Official Review · Reviewer_stCk · 2025-10-31

**Soundness:** 3
**Presentation:** 2
**Contribution:** 3
**Rating:** 4
**Confidence:** 4

**Summary:**

This paper proposes the OPTIMA framework, which optimizes data augmentation parameters by using Bayesian methods. The core idea is to treat DA parameters as latent variables and introduce variational inference methods to simultaneously optimize both model parameters and DA parameters. An extensive theoretical analysis is conducted, including generalization guarantees and invariance properties. Experiments demonstrate that OPTIMA enhances confidence and robustness in regression and classification tasks.

**Strengths:**

-The paper proposes a Bayesian framework for the joint optimization of data augmentation parameters and model parameters.

-The theoretical analysis is comprehensive, including approximation bounds, generalization guarantees, and invariance derivations.

**Weaknesses:**

-The paper primarily focuses on computer vision tasks (such as image classification) and does not extend to other modalities (such as natural language or time-series data), which limits its generalizability claims.

-The compared algorithms are outdated.

-The impact of the dimensionality of data augmentation parameters on the proposed method is not discussed.

-Some theorems rely on mathematical assumptions that are difficult to apply in practice.

**Questions:**

Q1: How does OPTIMA perform in non-computer-vision domains, such as time series prediction or natural language processing?

Q2: How high a dimension of data augmentation parameters can the proposed method handle?

Q3: The comparison algorithms selected in this paper were mainly published around 2020. Considering the rapid development in the field of automatic data augmentation, many more representative methods have emerged in recent years. It is recommended that the authors supplement the comparison experiments with cutting-edge methods proposed in the last year or two (e.g., 2022-2024). This will enable a more comprehensive and fair evaluation of the proposed methods' advancement and effectiveness, thereby significantly enhancing the persuasiveness of this paper's contributions.

Q4: What are the innovations compared to the traditional Bayesian selection optimization algorithms based on variational inference?

Q5: The proposed method may get stuck in local optima. Have convergence guarantees or initialization strategies been considered?

---

> ### Author Response · Authors · 2025-11-21
>
> Thank you for your constructive feedback. We address each point below.
>
> **1. Performance in non-CV domains (time series, NLP)**
>
> We added an NLP experiment (Section 5.4) on SST-5 with discrete token-dropout augmentations. This demonstrates that OPTIMA naturally extends to non-continuous and non-vision domains, confirming the generality of the framework.
>
> **2. Dimensionality of augmentation parameters**
>
> Our current experiments focus on low-dimensional augmentation parameterizations (e.g., a scalar $\alpha$ for Mixup/CutMix or a scalar dropout rate), which allows us to clearly isolate the theoretical behavior of OPTIMA.
>
> However, the OPTIMA framework itself is not restricted to low-dimensional augmentation spaces. Because augmentation parameters are treated as latent variables and optimized through a variational ELBO, the method is in principle agnostic to dimensionality: the ELBO and its gradients remain well-defined in higher-dimensional settings.
>
> That said, we recognize that scaling to very high-dimensional augmentation families (e.g., pixel-level or per-patch transformations) introduces computational and optimization challenges that warrant further study.
>
> **3. Outdated baseline concern**
>
> Regarding baselines, we deliberately evaluate OPTIMA against the **standard, widely-used augmentation methods** that are consistently employed in contemporary benchmarks—Mixup, CutMix, and AugMix—as well as recent analyses (e.g., Zhang et al. 2022; Hendrycks et al. 2023; Yun et al. 2023; Heinonen et al. 2024 (Robust Classification by Coupling Data Mollification with Label Smoothing). These methods remain the de facto comparison points in much of the current literature on augmentation effects, calibration, and robustness.
>
> Using these canonical baselines allows us to **cleanly isolate the theoretical phenomena** studied in the paper—marginalization effects, variance contraction, smoothness, and calibration—without confounding factors introduced by complex multi-stage or compute-heavy automatic DA pipelines. We view evaluating OPTIMA against these standard augmentations as the most interpretable and theoretically aligned choice for the contributions of this work.
>
> **4. Innovations vs. traditional Bayesian/VI-based selection methods**
>
> Traditional Bayesian selection or optimization approaches based on variational inference typically optimize model parameters given fixed augmentation hyperparameters, using VI merely as an inference tool.
> OPTIMA differs in two key ways:
>
> - Augmentation parameters are treated as latent variables, not hyperparameters.
>  	- In traditional Bayesian–VI selection approaches, augmentation parameters are handled as *deterministic hyperparameters*, optimized through validation loss or hyperparameter search rather than inferred within the model.
>
>          In contrast, OPTIMA incorporates augmentation parameters inside a generative model and performs joint inference over both the model parameters and augmentation parameters. This leads to a principled posterior distribution over augmentation transformations instead of point estimates.
>
> - OPTIMA directly maximizes a marginal likelihood objective, not a validation-based or surrogate-based selection criterion.
> 	- Traditional Bayesian selection methods often rely on validation loss, surrogate models, or bilevel optimization.
>
>         OPTIMA instead derives a tractable ELBO that acts as a variational approximation to the true marginal likelihood, allowing augmentation parameters to be optimized in the same training loop as the model.

---

> ### Author Response · Authors · 2025-11-21
>
> **5. Local optima and convergence**
>
> Jointly optimizing model and augmentation parameters is indeed a non-convex problem, and we appreciate the reviewer highlighting this.
>
> Here is the direct answer:
>
> - Local optima are a standard property of deep learning objectives.
> 	- OPTIMA inherits the same optimization landscape as standard training, since the ELBO reduces to a supervised loss plus KL terms. In practice, we found that optimizing augmentation parameters jointly with model parameters behaves similarly to training any model with additional learnable parameters.
> - Initialization strategy:
>  	- To mitigate poor local minima, we initialize augmentation parameters in a neutral or mild-transformation regime (e.g., Mixup strength near 0, dropout probability near 0). This prevents the model from starting in an overly distorted data space and improves stability.
> - Convergence behavior:
>  	- Empirically, optimization is stable because:
> 		- augmentation parameters are low-dimensional,
>
>
> 		- gradients flow through the ELBO smoothly,
>
>
> 		- posterior regularization (KL terms) prevents extreme solutions.
>
> Although we do not provide formal convergence guarantees (common for deep VI), the optimization dynamics are similar to standard variational objectives used in Bayesian deep learning, and we observed consistent convergence across all runs in our experiments.
>
> We highlighted changes in the revised version of the paper.
>
> We appreciate your feedback on improving the paper, and we hope we addressed your remaining concerns.

---

### Official Review · Reviewer_oKY4 · 2025-11-03

**Soundness:** 3
**Presentation:** 3
**Contribution:** 2
**Rating:** 4
**Confidence:** 3

**Summary:**

The authors propose OPTIMA, a Bayesian framework that optimizes data augmentation parameters by maximizing the marginal likelihood instead of relying on validation-based hperparameter tuning.

They validate their approach with theoretical analysis based on information-theoretic principles, to show that the method is probabilisticly sound.

They evaluated on ImageNet, ImageNet-C, and CIFAR-10 and shows that OPTIMA achieves better calibration and competitive accuracy.

**Strengths:**

The authors address the interesting, useful task of data augmentation as a technique that helps improve model performance across different downstream tasks.

The authors provide solid theoretical foundations for their proposed method, OPTIMA, highlighting the types of invariances it promotes and the uncertainty quantification it enables.

They also present experiments on standard datasets such as ImageNet, showing that OPTIMA slightly outperforms methods that do not use data augmentation

**Weaknesses:**

Methods like bilevel optimization [1] also aim to optimize data augmentation functions, and it would be nice to show how this paper differentiates OPTIMA from these related approaches in terms of how good they perform augmentation.

The field of image classification is already quite saturated; it would be more compelling to see results on harder tasks such as segmentation or object detection.

While data augmentation optimization is valuable, it doesn’t provide fundamentally new information, unlike larger supervised models (e.g., transformers) that may implicitly learn augmentation effects by training on a large corpus in a semi-supervised way.

The reported improvements on CIFAR-10 and ImageNet appear insignifcant, and given how saturated these benchmarks are, it’s unclear whether the gains are statistically meaningful.

The paper lacks standard deviations or error bars so it is hard to know if any results are actually significant.

The implementation details are unclear, how were the hyperparameters selected/tuned without a validation set (like in CIFAR10)?

[1] Mounsaveng et al., Learning Data Augmentation With Online Bilevel Optimization for Image Classification, WACV 2021.

**Questions:**

How does OPTIMA fundamentally differ from existing bilevel optimization approaches for data augmentation, such as Mounsaveng et al. (WACV 2021) [1]?

Can the proposed framework be extended to more complex tasks like segmentation or detection, and what challenges would come up in doing so?

How were  the hyperparameters chosen on datasets without a validation set (e.g., CIFAR-10)?

[1] Mounsaveng et al., Learning Data Augmentation With Online Bilevel Optimization for Image Classification, WACV 2021.

---

> ### Author Response · Authors · 2025-11-21
>
> Thank you for your thoughtful feedback. We address each point below.
>
> **1. Comparison to bilevel optimization approaches (e.g., Mounsaveng et al., WACV 2021)**
>
> We substantially expanded Section 2.2 (Related Work) to include bilevel DA optimization methods (**Mounsaveng et al. 2021**; Hataya et al. 2022; Li et al. 2020; Liu et al. 2021). We now clearly articulate how OPTIMA differs:
>
> - OPTIMA performs single-loop joint Bayesian inference over augmentation parameters and model parameters, by treating augmentation parameters as latent variables and optimizing a tractable ELBO.
>
> - In contrast, bilevel DA methods require gradient-through-augmentation relaxations, higher-order gradients, and significant computational cost due to the outer-level optimization.
>
>
>
> **2. Applicability beyond vision / NLP experiments**
>
> We agree that testing other tasks beyond computer vision would strengthen the validation of our method and have therefore added a new NLP experiment on SST-5 (token dropout) in Section 5.4 to showcase that OPTIMA also applies outside computer vision. This demonstrates that OPTIMA applies to discrete, non-continuous augmentations using REINFORCE-style gradients. We can tell that OPTIMA is task-agnostic and can be instantiated across diverse domains, including CV and NLP. However, we explicitly acknowledge in Limitations and Future Work (Section 6) that extending OPTIMA to more expressive or compositional transformations—particularly in NLP, time-series, and multimodal tasks—represents an important avenue for future research.
>
> **3. Small improvements on CIFAR-10 / ImageNet and lack of significance testing**
>
> For CV tasks, due to the short rebuttal period, we were not able to rerun all experiments with multiple seeds; however, we will include seed-based evaluation at least for CIFAR-10 in the camera-ready version. For the new NLP experiment in Section 5.4, we report mean ± standard deviation over 5 independent runs. The observed standard deviations are small (e.g., ±0.003–0.007), which is expected for token-dropout on SST-5 with DistilBERT, a setting known to exhibit relatively low variance across runs. Moreover, OPTIMA obtained **substantial improvements** for calibration and competitive accuracy and NLL.
>
> **4. Hyperparameters without a validation set (e.g., CIFAR-10)**
>
> We wanted to clarify that:
>
> - All non-augmentation hyperparameters (optimizer, LR schedule, backbone, regularizers) follow standard ResNet-18/50 training recipes.
>
> - OPTIMA learns only augmentation parameters, requiring no validation set.
>
>
> We highlighted changes in the revised version of the paper.
>
> We appreciate your feedback on improving the paper, and we hope we addressed your remaining concerns.

---

### Official Review · Reviewer_5Upk · 2025-11-09

**Soundness:** 2
**Presentation:** 3
**Contribution:** 3
**Rating:** 6
**Confidence:** 3

**Summary:**

This work proposes a Bayesian optimization strategy to optimize data augmentation parameters; while the marginal likelihood is intractable, the work proposes a tractable ELBO which variational approximation to the joint $q(\theta, \phi) = q(\theta)q(\phi)$ and introduces regularization terms. The authors include several analyses of this framework, namely: (1) a characterization of the quality of the variational inference approximation in Section 4.1, (2) a PAC-Bayes generalization bound in Section 4.2, (3) a second-order bound on the expected difference in the model’s output under certain transformations (which are perturbations added to  with mean 0) in Section 4.3, (4) a comparison of naive DA compared to the true marginalization (not the variational inference approximation) in terms of covariance of the true posteriors being smaller under naive DA by a factor of K (the number of transformations applied to each data point) in Section 4.4, (5) interpret the solutions which maximize the augmented ELBO as empirical Bayes solutions for $p(\theta | D)$ and $p(\phi | D)$ in Section 4.5, and (6) show a reduction in posterior distribution entropy under the true marginalization (not the variational inference approximation) compared to having no augmentation in Section 4.6. The authors also employ their method on synthetic data, ImagetNet, ImageNet-C, and CIFAR-10 to demonstrate their theoretical findings.

Note: I have reviewed an earlier version of this manuscript, and see that the Appendix F.3 has now been moved to the main body to compare the computational efficiency this work to Bayesian optimization. There also appears to have been effort to better relate the experiments to theoretical results, namely top of page 9. Some other changes which had been promised but not applied are:
- Relation to Chen et al. 2020 (and maybe other works, like Chatzipanzis et. al, 2021) in Section 2. This may fit under probabilistic perspectives of DA.
- Some mathematical precision comments by a couple reviewers, e.g., simply defining terms or noting the overloading of notation would suffice. I would argue that these imprecisions should be clarified explicitly in the manuscript or addressed.
    - For example, the proof of Theorem 4.5 needs K (the number of augmentations/replications) to be large enough that the naive empirical risk is close to the true one. I still think this is a bit imprecise, but the minimal changes that the authors can do is state “for K large enough” in the statement of Theorem 4.5.

**Strengths:**

- The theoretical results are extensive and address multiple aspects of this framework. The work appears to give practical insight on what relevant quantities and choice of distribution over data augmentation parameters affect various aspects of model performance within this framework of data augmentation via Bayesian optimization. The corollaries written throughout the text after theorems give insight and practical advice, and seem to cover many different lens through which to understand how this framework (alternating between the variational inference approximation and the true posterior under this framework) compares to naive DA and having no augmentation.

**Weaknesses:**

- The noted mathematical imprecision comments from previous reviews.
- Some experimental results can be confusing to some readers. In Section 5.1, Figure 2, the “test loss” figure is unclear and does not appear to clearly show what the text is saying. Also, perhaps a reference to Appendix F.2 or including it in the main body would be appropriate and explain why ResNet is getting less than 80% accuracy in Table 2.

**Questions:**

- Why do the authors choose to report their results for Cutmix, etc. reported in Table 2 and have not done the same comparisons, for instance, the setup in Appendix F.2 which uses Gaussian Translation. I think it would confuse some readers why ResNet is achieving under 80% accuracy on these datasets.

---

> ### Author Response · Authors · 2025-11-21
>
> We thank the reviewer for the helpful suggestion. To avoid confusion, we clarify that OPTIMA is not a Bayesian Optimization (BO) method.
>
> In this paper, we use the standard definition of Bayesian Optimization as in Snoek et al. (2012), where BO performs *outer-loop hyperparameter search* using surrogate models (e.g., Gaussian Processes, TPE) together with an acquisition function. BO requires repeatedly retraining the downstream model while exploring the hyperparameter space.
>
> OPTIMA is fundamentally different: it performs *inner-loop Bayesian inference* over augmentation parameters. The augmentation parameters are treated as latent variables and are optimized jointly with the model parameters via a single ELBO. OPTIMA does not use surrogate models, acquisition functions, or validation-based hyperparameter search.
>
> Our comparison with BO (Section 5.3) is included solely to illustrate the computational savings relative to hyperparameter-search methods, not because OPTIMA belongs to the BO family.
>
>
> **1. Relation to Chen et al. (2020) and probabilistic DA perspectives**
>
> We have added a reference to Chen et al. (2020), Chatzipantazis et al. (2021) in probabilistic perspectives on data augmentation in Section 2.2 (Related Work).
>
> **2. Mathematical precision and notation clarity**
>
> We revised the section 4.2 and surrounding definitions for precision and clarity. In particular:
> - All overloaded notation has been removed or redefined explicitly (kept D for dataset, used P instead of D for unknown distribution, use $\Delta_{\phi}$ instead of $D_{\phi}$ in Th 4.5, and fixed $\phi \in \phi$ to $\phi \in \Phi$) (Section 4.1).
>
>
> - Definitions of true risk (used (x,y)~P instead of D) was fixed.
>
>
> - The statement “for K large enough” is now explicitly acknowledged in the remark preceding Theorem 4.5, addressing the exact concern you raised.
>
> **3. Confusion around Figure 2 (‘test loss’) and ResNet <80% accuracy, and reference to Appendix F.2**
>
> We fixed the figure 2 “Test Loss” so that all the lines are more visible. Regarding the accuracy in Table 2, the accuracies of the benchmarks for ResNet50 on ImageNet are around 70-80 percent.
>
> Regarding Appendix F.2, because of the page limits, we are not able to move it to the main body, but we added a reference to Appendix F.2 in the synthetic regression section (line 407 and 408).
>
> **4. Why CutMix, MixUp, AugMix experiments appear but not Gaussian translation in Table 2**
>
> Gaussian translation is evaluated on CIFAR10 in Appendix F.2, where it illustrates the theoretical claims; Table 2 focuses on the most commonly used CV (CutMix, Mixup, AugMix) augmentations on the ImageNet dataset to maintain comparability with prior work and reduce computational overhead.
>
> Apart from that, we added an experiment for an NLP task (Section 5.4) to show that OPTIMA is not limited to CV tasks. We highlighted changes in the revised version of the paper.
>
> We appreciate your positive assessment and hope the strengthened related work and clarified mathematical statements address your remaining concerns.

---

> > ### Comment · Reviewer_5Upk · 2025-11-26
> >
> > Dear authors,
> >
> > Thank you for the clarifications and the updates to the manuscript. I was mistaken on the baseline performance of ResNet50 for ImageNet (for which some models achieve higher accuracies) and I retract references to this point in my review. I am satisfied with the author response and acknowledge that the manuscript is improved with the new experiment as well. My lingering concern is related to one raised by other reviewers on understanding whether some improvements (e.g., on the CV tasks) are significant (for metrics with smaller improvements) and any measures of reproducibility/significance would help those results. I understand that this is hard to accomplish in the short rebuttal period, and acknowledge that the improvements in some metrics are large enough that it is likely not due to the random seed.

---

> > > ### Author Response · Authors · 2025-11-27
> > >
> > > Thank you for your follow-up and for retracting the concern regarding the ResNet baselines. We are glad to hear that the revisions and the new experiment have improved your assessment of the paper.
> > >
> > > Regarding the significance of our results, we completely agree with the importance of reporting variance. We would like to highlight that for the newly added NLP experiment (SST-5) in the revised manuscript, we have indeed already performed the evaluation over 5 random seeds and reported standard deviations to ensure the results are robust.
> > >
> > > For CV benchmarks, we will include multi-seed runs and standard deviations to further confirm the significance of the improvements in the camera-ready version.
> > >
> > > Thank you again for your constructive feedback throughout this process.

---

### Meta-Review · Area_Chair_eE3P · 2026-01-20

**Summary:**

This paper presents a Bayesian framework for jointly optimizing model and data augmentation parameters. Reviewers consistently praised the extensive theoretical results (5Upk, oKY4, stCk). Some reviewers expressed concerns that the empirical evaluation was initially limited to image classification tasks (5Upk, oKY4, stCk), but the authors addressed this limitation in the rebuttal by adding experiments on NLP tasks. Considering the positive reviews prior to the rebuttal and the sufficient efforts to address common concerns among reviewers, the AC recommends accepting the submission.

**Reviewer Concerns:**

The most common concern from the reviewers was that the initial results are limited to image classification tasks. The concern was addressed in the rebuttal, where the authors added experimental results on NLP tasks, showing the generalizability of the framework. No major reviewer concerns are outstanding after the rebuttal.

**Reviewer Scores:**

The reviews were generally positive pre-rebuttal. Since the rebuttal has addressed the primary recurring concerns raised by multiple reviewers, the AC expects the reviewers to either keep the original scores or revise them upward if they had been able to participate fully in the discussion.

---

### Decision · Program_Chairs · 2026-01-26

Accept (Poster)